# 3D POINT CLOUD SEQUENCES AS 2D VIDEOS

## ABSTRACT

The irregular and unstructured nature of 3D point cloud sequences in both spatial and temporal domains poses great difficulties in extracting their discriminative features effectively and efficiently. To tackle these challenges, in contrast to existing methods devoted to developing special architectures for modeling sequences, we advocate a new paradigm by proposing a novel representation modality, called *point geometry video* (PGV), that encodes the coordinates of the 3D points of a sequence as the pixel values of a 2D color video, with the original spatial neighborhood relationship and temporal consistency preserved. PGV significantly facilitates the processing of sequential 3D point clouds by enabling the adaption of powerful learning techniques for 2D image and video processing. Technically, by leveraging the local aggregation and kernel-sharing properties of the convolution operation, we build a self-supervised auto-encoder composed of convolutional layers, that consumes pre-defined regular grids to produce the PGV representation of a sequence of point clouds. We demonstrate the superiority and generality of the PGV on downstream tasks, including sequence correspondence, spatial upsampling, and forecasting. The PGV as a novel representation modality opens up new possibilities for deep learning-based processing and analysis of point cloud sequences. The code and data will be made publicly available.

## 1 INTRODUCTION

A 3D point cloud sequence is a series of 3D point cloud frames taken at consecutive time steps, depicting the motion/change of an object/scene. Different from 2D video signals that are defined on regular dense grids and thus can be processed by powerful 2D/3D convolutional neural networks (CNNs) (Kalluri et al., 2023; Shi et al., 2022; Gao et al., 2022), 3D point cloud sequences are characterized by the lack of a regular structure in both spatially and temporally, which poses great challenges in developing efficient learning architectures for modeling such data. The lack of correspondence across consecutive frames of point clouds makes it difficult to achieve reliable temporal modeling. As a consequence, spatial-temporal neighborhood construction and down-sampling become non-trivial and computationally intensive. Therefore, for many generative and reconstructive learning tasks, one can only uses those distribution-based losses (such as the earth mover's distance (EMD) and Chamfer distance (CD)), which are either inefficient or ineffective (Nguyen et al., 2021; Wu et al., 2021).

Currently, the research on deep learning-based point cloud sequence processing is still in the early stage. Most of the existing research is focused on developing specialized learning mechanisms and processing modules to cope with the unstructured characteristics of sequential point data. For example, existing works seek to aggregate features, use voxel (Wang et al., 2020) or implicit (Lei & Daniilidis, 2022; Mihajlovic et al., 2021) representations, or focus on the correspondence problem for paired point clouds (Zeng et al., 2021) or meshes (Donati et al., 2020; Eisenberger et al., 2021; Donati et al., 2022). These works either require geometry priors (e.g., geodesics) or a sizeable dataset to achieve generalization and suffer from limited performance when handling sequences with significant non-rigid motion or deformation.

Unlike existing works, *we advocate a new research paradigm*, i.e., developing a unified, explicit, and structured representation modality for the general processing of point cloud sequences. **Conceptually**, as illustrated in Fig. 1, we propose to represent an arbitrary 3D point cloud sequence with *unknown* temporal correspondence as a 2D color video, called the *point geometry video* (PGV). Intuitively, as a data structurization procedure, the actual learning process of producing the PGV can

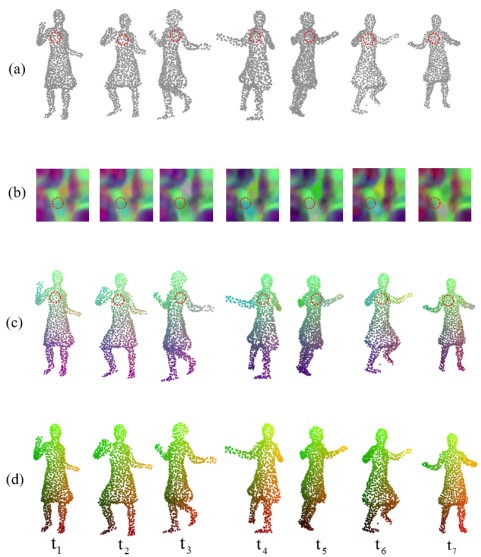

Figure 1: (**a**) A 3D point cloud sequence with point-wise correspondence across frames ***unknown***. (**b**) The PGV representation of the above sequence, where each 2D color image corresponding to a point cloud frame is derived by encoding the coordinates (i.e., $[x, y, z]$) of spatially *neighboring* 3D points as the pixel values (i.e., $[r, g, b]$) of *adjacent* pixel locations, (***spatial smoothness***); moreover, the pixels with the same location across frames generally correspond to an identical position of the 3D object (***temporal consistency***). (**c**) Visualization of the above PGV in 3D space, where the value of a pixel is the coordinate of a 3D point rendered as the same color as the corresponding pixel, demonstrating the spatial smoothness property. (**d**) Visualization of the above PGV in 3D space, where the PGV pixels with an identical location across frames are rendered with an identical color, demonstrating the temporal consistency property. Note that the circled regions in different representations are corresponded.

be interpreted as mapping spatio-temporal 3D points to appropriate 2D pixel locations of a series of 2D image grids with desired *spatial smoothness* and *temporal consistency*, followed by encoding the 3D coordinates as color pixel values. Therefore, the *generic* PGV representation modality can serve as a *bridge* to facilitate direct application of established powerful learning techniques and network design for 2D images or videos to boost the processing and analysis of point cloud sequences.

**Technically**, leveraging the local aggregation and kernel-sharing properties of the convolution operation, we developed a convolution-based auto-encoder framework shown in Fig. 2 to produce the PGV representation in a *self-supervised* and *over-fitting* manner. We develop the framework in a coarse-to-fine fashion. Specifically, for each frame of point ckouds of a given sequence, we regress a 2D color image with spatial smoothness from a pre-defined 2D regular grid. Guided by the resulting 2D color images, we further simultaneously deform the spatial and temporal structures of a pre-defined 3D regular grid to ensure both spatial smoothness and temporal consistency of the produced PGV. This framework is trained with the EMD loss between input sequence and the point cloud output by network. We demonstrate the superiority and generality of the point geometry video (PGV) by developing several PGV-based pipelines for efficient 3D point cloud sequence processing, which achieve impressive performance.

In summary, the main contributions of this paper are:

- a new regular representation modality for 3D point cloud sequences, namely PGV;
- a self-supervised learning framework for generating PGVs; and
- state-of-the-art baselines for temporal correspondence, spatial upsampling, and future frame synthesis of 3D point cloud sequences.

## 2 RELATED WORK

**Deep Learning on 3D Point Clouds**. Recent years have witnessed a proliferation of deep set architectures directly operating on 3D point clouds, as pioneered by Qi et al. (2017a;b). The follow-up works further investigated various more advanced point convolution-based (Li et al., 2018a;b; Thomas et al., 2019; Liu et al., 2019c; Xu et al., 2021b; Xiang et al., 2021), graph-based (Verma et al., 2018; Wang et al., 2019; Xu et al., 2020), and transformer-based (Guo et al., 2021; Zhao et al., 2021; Park et al., 2022) modeling architectures. However, these approaches are developed for the single point cloud input, and the adaptation to point cloud sequence processing is still highly non-trivial due to the difficulty and complexity of spatio-temporal geometric modeling.

Another line of work resorts to a different perspective for overcoming the irregularity and unstructuredness of 3D point clouds by creating regular 2D geometry image (GI) (Gu et al., 2002) or GI-like representation structures, on which various mature 2D learning architectures (e.g., convolutional neural networks (CNNs)) and processing techniques can be directly applied. Representatively, Sinha et al. (2016; 2017); Maron et al. (2017); Haim et al. (2019) exploited mesh parameterization techniques to implement surface-to-plane mapping, while Zhang et al. (2022; 2023) proposed to directly learn deep regular representations from point clouds. These works are also related to a thread of point cloud decoders that deform a pre-defined pattern, such as a 2D grid (Yang et al.,

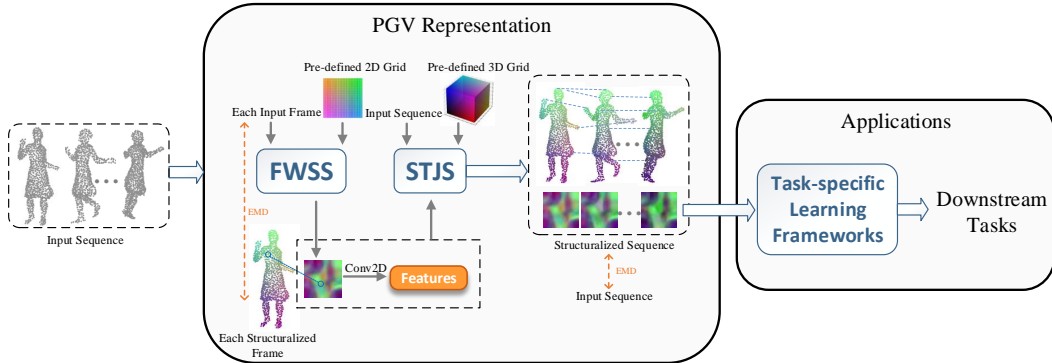

Figure 2: Flowchart of representing a point cloud sequence as a PGV and its applications. **FWSS** and **STJS** refer to the *frame-wise spatial structurization* and *spatio-temporal joint structurization* modules, respectively. The PGV representation stage (Sec. 3) operates sequence-by-sequence in an *over-fitting* manner. The applications stage takes the PGV as input for downstream tasks (Sec. 4).

2018; Groueix et al., 2018; Zhao et al., 2019; Pang et al., 2021). It is important to note that these works encode the input point cloud into a latent codeword, which technically differs from the over-fitting approach of our PGV representation. Moreover, the aforementioned approaches are limited to static geometric representations and are, therefore, inapplicable to dynamic point cloud sequences requiring spatio-temporal structurization.

**Explicit vs. Implicit Learning on 3D Point Cloud Sequences.** As a pioneering work learning on point cloud sequences explicitly, Liu et al. (2019b) focused on the motion range of an object and proposed two methods for determining the spatio-temporal neighborhood: direct grouping and chain flow grouping. The subsequent works made efforts to design various convolutional and transformer-based learning structures for raw point cloud sequence modeling, where the spatial and temporal domains can be modeled in a decoupled (Fan et al., 2021b;c) or joint (Fan et al., 2021a; 2022; Wei et al., 2022) learning manner. However, these studies didn't fully address the irregularities in 3D point cloud sequences, especially the correspondence in the time domain. Li et al. (2021b) learned the underlying temporal coherence in point cloud sequences implicitly. Zhong et al. (2022) extended the kinematic concept of ST-surfaces to the feature space, enabling the capture of 3D motions without explicit tracking. He et al. (2022) proposed estimating frame-wise motions of points across different frames through sequential scene flow estimation using spatio-temporal interactions. However, managing point cloud sequences showcasing significant deformation remains a challenge. In contrast to the previously mentioned approaches, our work markes an early exploration into structuralizing point cloud sequences explicitly.

Another research trajectory delves into implicit representation methods. Niemeyer et al. (2019) utilized implicit strategies for 4D reconstruction, leveraging ground truth occupancy and trajectory. Rempe et al. (2020) presented spatio-temporal representations for objects in normalized coordinate space. Lei & Daniilidis (2022) addressed inter-frame deformations using continuous bijective canonical maps centered on a canonical shape. Wang et al. (2020) extracted voxels directly from depth videos for 3D motion encoding, which has potential implications for storage consumption and shape distortion. Many of the aforementioned works rely on mesh, voxel, or occupancy rather than pure point cloud sequences. In contrast to these works, our paper focuses on self-supervised explicit representations, and we explicitly parameterize the point cloud sequence as PGV, ensuring a compact representation that minimizes distortion and storage concerns.

## 3 PROPOSED METHOD

**Problem Statement.** Let $\{\mathbf{P}_t \in \mathbb{R}^{N \times 3}\}_{t=0}^{T-1}$ be an arbitrary 3D point cloud sequence with $T$ frames each of $N$ points[1], where point-wise correspondence across frames is *unknown* and $\mathbf{p}_t^i := \left[x_t^i, y_t^i, z_t^i\right] \in \mathbb{R}^{1 \times 3}$ is a 3D point of frame $t$. Our objective is to represent $\{\mathbf{P}_t\}_{t=0}^{T-1}$ as a 2D colored video with $T$ frames of dimensions $H \times W$ ($H = W = \sqrt{N}$), namely point geometry video (PGV), denoted as $\{\mathbf{G}_t \in \mathbb{R}^{H \times W \times 3}\}_{t=0}^{T-1}$. More specifically, we attempt to re-organize and encode points of $\{\mathbf{P}_t\}_{t=0}^{T-1}$ into appropriate pixel locations of $\{\mathbf{G}_t\}_{t=0}^{T-1}$, i.e., $\mathbf{G}_t(u,v,1) = x_t^i$, $\mathbf{G}_t(u,v,2) = y_t^i$, and $\mathbf{G}_t(u,v,3) = z_t^i$, where $(u,v)$ is the 2D pixel coordinate ($1 \leq u, \ v \leq \sqrt{N}$). At the same time, the

---

[1]Note that the points of $\mathbf{P}_t$ are stacked randomly to form a matrix.

resulting PGV should have the following two characteristics, which are necessary in order to adapt PGVs to the working mechanism of 2D and 3D convolution operations (i.e., neighbor aggregation):

- *Spatial Smoothness*: the pixels in a local patch of $\mathbf{G}_t$ correspond to a set of neighboring points in 3D space; and
- *Temporal Consistency*: the pixels with identical $(u,\ v)$ across all $T$ frames are corresponded, i.e., they correspond to the same position of the object/scene.

**Technical Motivation**. The above-mentioned problem theoretically refers to 3D surface parameterization (Sheffer et al., 2006). Intuitively, one can construct a neural network, denoted as $f_{\boldsymbol{\theta}}(\cdot)$, which can consume 3D points to predict their 2D pixel coordinates, i.e., $\{(u_t^i, v_t^i)\}_{i=1}^N = f_{\boldsymbol{\theta}}(\{\mathbf{p}_t^i\}_{i=1}^N)$, and then encode the 3D coordinates as 2D pixel values. However, for point clouds with arbitrary and non-manifold geometric structures, it is challenging or even infeasible to obtain the ground-truth 2D coordinates that are necessary for supervising the training of $f_{\boldsymbol{\theta}}(\cdot)$. Moreover, the difficulty is further amplified when parameterizing a series of point cloud frames with the temporal consistency required.

In view of the above-mentioned difficulty, we consider a self-supervised framework from the perspective of self-reconstruction. Generally, we construct a network composed of 2D convolution (Conv2D) $f_{\boldsymbol{\psi}}(\cdot)$, which deforms a pre-defined 2D regular grid[2] $\mathbf{C} \in \mathbb{R}^{H \times W \times 3}$ to over-fit each of $\{\mathbf{P}_t\}_{t=0}^{T-1}$ independently as a 2D color image $\mathbf{I}_t = f_{\boldsymbol{\psi}}(\mathbf{C})$. The learning process of $f_{\boldsymbol{\psi}}(\cdot)$ is driven by minimizing the EMD loss between $\mathbf{P}_t$ and the point cloud re-organized from $\mathbf{I}_t$. The *intention* of such an approach lies in that owing to the local aggregation and kernel sharing properties of Conv2D, the network tends to progressively modify the input in a smooth manner, thus retaining and propagating the spatial smoothness of $\mathbf{C}$ to $\mathbf{I}_t$ to some extent, while the EMD can promote the underlying shape of $\mathbf{I}_t$ to approach that of $\mathbf{P}_t$. Furthermore, the above fashion can be extended to sequences by employing a network of 3D convolution (Conv3D) $h_{\boldsymbol{\phi}}(\cdot)$ fed with a pre-defined regular 3D grid $\mathbf{Q} \in \mathbb{R}^{T \times H \times W \times 3}$ to take both spatial smoothness and temporal consistency into account, i.e., $\{\widetilde{\mathbf{G}}_t\}_{t=0}^{T-1} = h_{\boldsymbol{\phi}}(\mathbf{Q})$. Similarly, the learning of $h_{\boldsymbol{\phi}}(\cdot)$ can be driven by minimizing the EMD loss between $\{\mathbf{P}_t\}_{t=0}^{T-1}$ and the re-organized point cloud sequence from $\{\widetilde{\mathbf{G}}_t\}_{t=0}^{T-1}$.

**Framework Overview.** Instead of directly fitting a sequence from a pre-defined 3D regular grid via Conv3D, which produces poor representation quality (see the last column of Table 2), due to the great difficulty of jointly optimizing high-dimensional spatial and temporal structures from scratch, we construct a self-supervised framework in a coarse-to-fine fashion, as shown in Fig. 2, consisting of two modules: *frame-wise spatial structurization* in Sec. 3.1, which takes as input a pre-defined 2D regular grid to represent each point cloud frame of $\{\mathbf{P}_t\}_{t=0}^{T-1}$ as a 2D color image frame-by-frame in an over-fitting and multi-scale manner, and *spatio-temporal joint structurization* in Sec. 3.2, which simultaneously deforms the spatial and temporal structures of a pre-defined 3D regular grid, guided by the stack of the resulting 2D color images by the preceding module. As a generic representation modality, PGV enables the use of well-established learning techniques for 2D images/videos to construct 3D point cloud sequence processing pipelines demonstrated in Sec. 4.

***Remark***. It is worth noting that our representation modality is *different* from that by the conventional perspective/parallel projection suffering from occlusions, and our framework is *permutation-invariant* (i.e., irrelevant to the order of points). In addition, the regular structure of PGVs allows the utilization of pixel-wise $\ell_1$ and $\ell_2$ losses, which demonstrate the stronger supervision ability and are easier to optimize than the EMD and CD (Zeng et al., 2021; Feng et al., 2021).

## 3.1 FRAME-WISE SPATIAL STRUCTURIZATION

Building upon the aforementioned insight, the goal of this module is to independently represent each of $\{\mathbf{P}_t\}_{t=0}^{T-1}$ as a 2D color image $\mathbf{I}_t$. Architecturally, as shown in Fig. 3, it is composed of three components, i.e., feature encoder, hierarchically structured embedding $f_{\boldsymbol{\psi}}^e(\cdot)$ and regular frame extraction $f_{\boldsymbol{\psi}}^d(\cdot)$, all with Conv2D blocks.

Specifically, $f_{\boldsymbol{\psi}}^e(\cdot)$ maps an encoded down-scaled pre-defined 2D regular grid $\widetilde{\mathbf{C}} \in \mathbb{R}^{\frac{H}{4} \times \frac{W}{4} \times 3}$ into three embeddings at different scales $\{\mathbf{L}_l\}_{l=1}^3$, i.e., $\mathbf{L}_1 = f_{\boldsymbol{\psi}}^{e^2}(f_{\boldsymbol{\psi}}^{e^1}(\widetilde{\mathbf{C}}))$ and $\mathbf{L}_l = f_{\boldsymbol{\psi}}^{e^{l+1}}(\mathrm{UP}(\mathbf{L}_{l-1}))$

---

[2]The values of a pre-defined regular 2D (resp. 3D) grid are obtained by regularly sampling 3D coordinates within the first slice (resp. the entire volume) of the unit 3D cube $[0,\ 1]^3$.

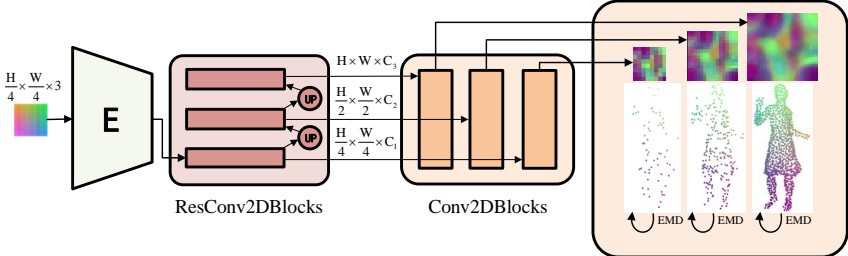

Figure 3: Flowchart of the frame-wise spatial structurization (**FWSS**) module in Sec. 3.1. "**UP**" denotes up-scaling. The input grid is passed into the encoder ("**E**") and a series of Conv2D-based blocks to get the hierarchically structured embedding, which will be converted to regular frames, driven by the EMDs at three different scales.

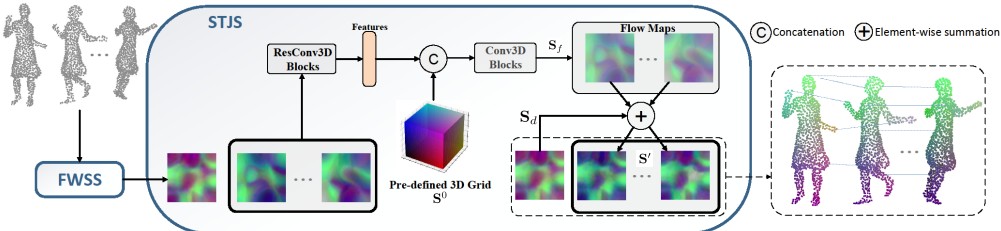

Figure 4: Flowchart of the spatio-temporal joint structurization (**STJS**) module. See the detailed description in 3.2.

for $l = 2$, $3$, where $f_{\psi}^{e^l}(\cdot)$ is the $l$-th Conv2D block and $\text{UP}(\cdot)$ is the up-scaling operation. Then, $f_{\psi}^{d}(\cdot)$ maps the multi-scale embeddings back to the original grid space to produce spatially smooth 2D color images of three scales $\{\mathbf{I}_t^l = f_{\psi}^{d^l}(\mathbf{L}_l)\}_{l=1}^3$, where $f_{\psi}^{d^l}(\cdot)$ is the block at the $l$-th scale. Finally, we set $\mathbf{I}_t^3$ of sizes $H \times W$ as $\mathbf{I}_t$.

We perform this module frame by frame in an over-fitting manner and drive its learning by minimizing the EMD loss between the down-scaled $\mathbf{P}_t$ and the point cloud re-organized from $\{\mathbf{I}_t^l\}_{l=1}^3$ at all three scales. See *Appendix* Sec. A.3 for ablation studies on loss functions and multi-scale strategy.

## 3.2 Spatio-Temporal Joint Structurization

By taking advantage of the local aggregation and kernel-sharing properties of Conv3D, this module learns to simultaneously optimize the spatial and temporal structures of a pre-defined 3D regular grid for pursuing the PGV with both spatial smoothness and temporal consistency. More importantly, we utilize the coarse information of the 2D color images by the preceding module, which are spatially smooth but without temporal consistency, to relieve the learning difficulty in the high-dimensional spatio-temporal space. Architecturally, as illustrated in Fig. 4, we implement this module as a 3D convolution-based geometry embedding framework, consisting of two components: the encoder $h_{\phi}^{e}(\cdot)$ in a residual learning fashion and the decoder $h_{\phi}^{d}(\cdot)$. Both $h_{\phi}^{e}(\cdot)$ and $h_{\phi}^{d}(\cdot)$ are equipped with Conv3D blocks.

Specifically, with $\{\mathbf{I}_t\}_{t=0}^{T-1}$ available, we set its first frame $\mathbf{I}_0$ as the beginning of the final PGV, i.e., $\mathbf{G}_0 = \mathbf{I}_0$. Then, we create $T-1$ copies of $\mathbf{I}_0$ and stack them along the time dimension to form $\mathbf{S}_d \in \mathbb{R}^{(T-1) \times H \times W \times 3}$. Similarly, we also stack the remaining frames $\{\mathbf{I}_t\}_{t=1}^{T-1}$ to form $\mathbf{S}_r \in \mathbb{R}^{(T-1) \times H \times W \times 3}$. Denote by $\mathbf{S}^0 \in \mathbb{R}^{(T-1) \times H \times W \times 3}$ a pre-defined 3D regular grid. We first learn the flow maps from $\mathbf{I}_0$ to each of $\{\mathbf{I}_t\}_{t=1}^{T-1}$ as $\mathbf{S}_f = h_{\phi}^{d}(\text{CAT}(h_{\phi}^{e}(\mathbf{S}_r), \mathbf{S}^0))$, where $\text{CAT}(\cdot)$ refers to the concatenation operation. And the resulting flow maps $\mathbf{S}_f \in \mathbb{R}^{(T-1) \times H \times W \times 3}$ is added to $\mathbf{S}_d$, leading to $\mathbf{S}' \in \mathbb{R}^{(T-1) \times H \times W \times 3}$, the stack of the remaining frames of the final PGV.

We perform this module in an *over-fitting* manner and drive its learning by minimizing the EMD loss between $\{\mathbf{P}_t\}_{t=0}^{T-1}$ and the point cloud sequence re-organized from the output PGV.

## 4 PGV-based Point Cloud Sequence Processing

To demonstrate the advantages of our PGV representation modality, we investigate three PGV-based sequence processing tasks. Note that to demonstrate the PGV's generality, we construct the frame-

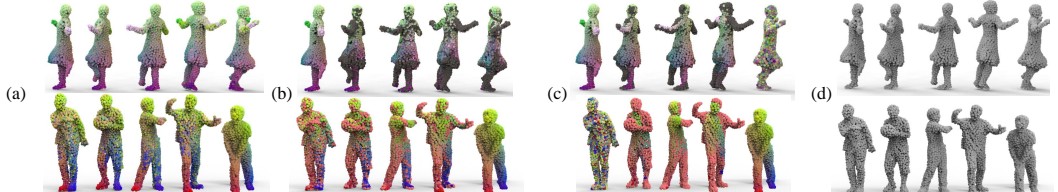

Figure 5: Visual comparison of the predicted dense correspondence. **Upper**: a natural sequence. **Bottom**: a pseudo sequence composed of five different subjects. (**a**) Ours; (**b**) CorrNet3D (Zeng et al., 2021); (**c**) Neuromorph (Eisenberger et al., 2021); (**d**) The input sequence. The colors indicate the temporal correspondence. Note that the **black** points in (b) and (c) mean that *no valid* correspondence is predicted from them. We refer readers to the *Appendix* Sec. A.4 and *video demo* for more visual results.

work for each task by employing existing deep models for 2D videos/images. More advanced deep models could be adopted for pursuing higher performance.

**Prediction of dense correspondence across frames of raw sequences** is a fundamental and challenging problem, which aims to align semantically corresponded points of any two frames in pointwise, given a raw sequence. Although many methods for dense 3D shape correspondence have been proposed (Deng et al., 2022), there is a very limited number of works directly handling a sequence of multiple point cloud frames. Moreover, most methods require ground-truth dense correspondence as supervision, limiting their practicality. As aforementioned, the constructed framework can represent raw sequences into PGVs with temporal consistency across frames promoted. Thus, our framework can be used to align raw sequences in an *self-supervised* fashion. Also, it is worth noting that in addition to *natural* sequences (i.e., the real motion/change of objects/scenes), this framework can also handle *pseudo* sequences, i.e., sequences constructed by concatenating point clouds of different objects.

**Spatial upsampling of point cloud sequences** aims to consistently densify all point cloud frames of an input spatially sparse sequence. Compared with single point cloud upsampling, it is more challenging due to problem of "how to effectively and efficiently explore the complementary information across different point cloud frames." Based on the spatial super-resolution model for multi-view images, named LFSSR-ATO (Jin et al., 2020), we construct an effective framework for sequence spatial upsampling, by taking each frame of the resulting PGV as a view. Generally, LFSSR-ATO is made up of an All-to-one SR module that performs individual super-resolution for each view of a low-frequency image by combining embedded data from all other views, followed by a structurally consistent regularization module that forces a low-frequency parallax structure in the reconstructed low-frequency image.

**Future point cloud frame synthesis**, similar to 2D video prediction, aims to generate future frames of a point cloud sequence based on preceding frames, which is challenging, especially for sequences with large non-rigid deformation. Credited to the regular structure of our PGV representation, we employ two recent video processing models, i.e., FLAVR (Kalluri et al., 2023) and SimVP (Gao et al., 2022), to achieve this task. Generally, FLAVR utilizes 3D convolution and deconvolution for an efficient auto-encoder design, while SimVP relies on 2D convolutional modules.

We refer readers to the *Appendix* Sec. A.1 for the detailed frameworks of the latter two tasks. Note that the latter two downstream processing tasks are performed under the train-test setting, i.e., the sequences in the form of PGVs are split into training and testing sets.

## 5 EXPERIMENTS

We experimented with three downstream tasks on 16 sequences obtained from MIT-Animation (Vlasic et al., 2008) and 8iVSLF (Krivokuca et al., 2018), where the training set consists of 8 sequences and the other 8 sequences are used for testing. We applied FPS (Eldar et al., 1997) to uniformly sample 1024 points for each 3D point cloud frame in our sequence correspondence and forecasting tasks. For our sequence upsampling task, we sampled 2500 points for the *Swing* sequence, which can be randomly separated into sub-groups each with eight frames. *Note that our representation is agnostic to specific object types, as no modules are tailored to any particular category, making it generalizable to various object types.*

### 5.1 DENSE CORRESPONDENCE PREDICTION

We compared with two recent unsupervised correspondence learning methods named CorrNet3D (Zeng et al., 2021) and Neuromorph (Eisenberger et al., 2021). Note that CorrNet3D only

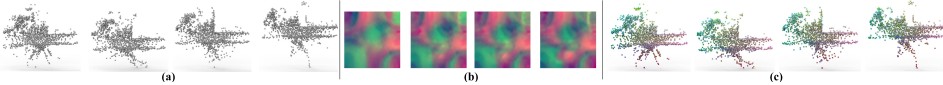

Figure 7: Visual comparison of the results by different methods used for spatially upsampling of sequences. (**a**) Input sparse sequence (**b**) Ground-truth dense sequence (**c**) Neural Points (Feng et al., 2022) (**d**) MAFU (Qian et al., 2021) (**e**) Ours.

Figure 8: Visualizations on the sequence '04' of SemanticKITTI (Behley et al., 2019) dataset. (**a**) visualizes the input sequences with **unknown** dense correspondence across frames. (**b**) visualizes the resulting PGV representation. (**c**) visualizes the resulting PGV in 3D space, where the colors indicate dense correspondence across frames.

handles pairwise point clouds, and Neuromorph requires the geometry prior (i.e., geodesic information) provided by ground-truth mesh models. The GT correspondence is unavailable for the input sequences of all methods. To perform quantitative evaluation, for each pair of adjacent frames of the regular sequence, we calculated the corresponding proportion of the $k$-nearest neighborhood of each point, and proposed an algorithm to calculate the correspondence ratio in percentage, considering both the reconstruction and correspondence accuracy. We refer readers to the *Appendix* Sec. A.2 for the pseudocode of the algorithm.

**Results of Natural Sequences.** We applied the pre-trained model of CorrNet3D (Zeng et al., 2021) on paired human models and re-trained Neuromorph (Eisenberger et al., 2021) using paired data deduced from the training set. We fed 8 successive frames into each model during testing to infer correspondence. Note that these two approaches slide from frame to frame for the generation of sequence correspondence, while our approach can concurrently fit the entire sequence to provide a structured PGV with higher efficacy. We reported the quantitative results under different numbers of neighbors $k$ on the *Swing* sequence in Fig. 6 (upper), where our method provides more reliable correspondences than the other point-based and mesh-based methods. Typical visual examples are given in the upper row of Fig. 5. When dealing with sequences with non-rigid motion, our method successfully deduces a regular sequence with naturally-preserved dense correspondences. Additionally, Fig. 8 showcases our method's capability in handling rigid motions, producing the PGV representation of a LiDAR sequence with coherent correspondence.

**Results of Pseudo Sequences.** We presented the quantitative indicators in Fig. 6 (bottom). Given a highly challenging input sequence, our method can still achieve more than 28% correspondence ratio, which is over 1.5 times higher than the competing methods. We also provided the visual examples in the bottom row of Fig. 5, composed of mixed sequences starting from right to left. Note that Neuromorph learns correspondence through an interpolation process without generating correspondence for the last frame, while our method can handle the whole sequence without such limitation. Moreover, our method reaches a larger proportion of correspondence than other methods. For the difficult case of the "basketball" component, our method can still produce a plausible correspondence near the hands across other shapes.

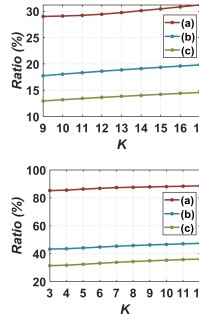

Figure 6: Quantitative results. **Upper**: the *Swing* sequence. **Bottom**: a pseudo sequence composed of the shapes of five different subjects. (a) Ours (b) CorrNet3D (Zeng et al., 2021) (c) Neuromorph (Eisenberger et al., 2021).

## 5.2 SPATIAL UP-SAMPLING FOR SEQUENCES

We compared the baseline method in Sec. 4 with state-of-the-art point cloud up-sampling methods[3], i.e., MAFU (Qian et al., 2021) and Neural Points (Feng et al., 2022), which spatially up-sample a sequence frame by frame. We adopted EMD and mNUC (Li et al., 2019) metrics for quantitative evaluation.

---

[3]We attempted to compare with the only two works for upsampling point cloud sequences (Wang et al., 2021; 2022). However, their codes are not publicly available, and the authors declined our request for codes, making us unable to reproduce reasonable results, only based on the details given in the papers.

Table 1: Quantitative evaluation ($\times 10^{-3}$) of various methods on spatial upsampling of sequences (columns 2-4) and future point cloud frames synthesis (columns 5-6).

| | Ours | NeuralPoints (Feng et al., 2022) | MAFU (Qian et al., 2021) | Ours (SimVP/FLAVR) | P4DTrans (Fan et al., 2021a) |
|---|---|---|---|---|---|
| EMD↓ | **1.335** | 8.887 | 4.903 | 37.560/**33.574** | 67.157 |
| mNUC↓ | **0.689** | 1.080 | 1.085 | 1.681/**1.352** | 2.379 |

| (a) | (b) | (c) | (d) | (e) |

Figure 9: Visual comparison of the results on future point cloud frame synthesis. The first five frames (**a**) were used as the input to predict the last three frames (**b**) of the ground-truth frames. Predicted results by (**c**) Ours (SimVP), (**d**) Ours (FLAVR), and (**e**) P4DTrans (Fan et al., 2021a).

Table 1 (columns 2-4) lists the quantitative results, where it can be seen that our baseline achieves a significantly lower EMD value while maintaining a lower uniformity cost. The superiority of our method in terms of reconstruction quality and uniformity is also validated in visual results provided in Fig. 7. These advantages of our baseline credited to (1) the adoption of mature 2D learning techniques (e.g., feature fusion, the $\ell_1$ loss, etc.) for powerful feature learning; and (2) the exploration of complementary information between frames, which are fundamentally enabled by our PGV representation modality.

## 5.3 FUTURE POINT CLOUD FRAME SYNTHESIS

To verify the potential of our PGV representation on this task scenario, we compared our baseline method (as introduced in Sec. 4) with P4DTrans (Fan et al., 2021a), a transformer-style architecture designed to avoid point tracking when capturing spatio-temporal correlation across the entire point cloud video. In practice, we fed five frames to all models to predict the subsequent three frames. We used the EMD loss for the supervision of P4DTrans, and the L1 loss for our video baseline.

As listed in Table 1 (columns 5-6), both of our baselines numerically exceed P4DTrans to a significant extent. This is because the structured PGV has naturally maintained spatial smoothness and temporal consistency, conducive to the learning of spatio-temporal information of 3D convolutional models. Besides, Fig. 9 provides visual comparisons, where it can be seen that the results produced by P4DTrans (Fig. 9e) cannot convey effective information, while our baseline methods produce more reasonable prediction results (Figs. 9c and 9d).

## 5.4 ABLATION STUDY

We performed necessary ablative experiments on the *Swing* sequence to understand our representation framework better, including its capability of key modules and performance under various spatial and temporal resolutions and lengths of the sequence.

Table 2: Ablation on the capability of the two modules FWSS in Fig. 3 and STJS in Fig. 4.

| Modules | FWSS-only | FWSS+STJS | STJS-only |
|---|---|---|---|
| Ratio | 25.45% | 86.12% | 58.13% |

**Effectiveness of Key Modules.** We omitted the spatio-temporal joint structurization (STJS) module and directly stacked the 2D color images by the frame-wise spatial structurization (FWSS) module as the resulting PGV representation. We reported quantitative results in Table 2, from which we can conclude that the absence of the STJS module significantly degrades the learned correspondence. As visualized in Fig. 11, the FWSS module can generate 2D color images with spatial smoothness, but lack temporal consistency (Fig. 11a), especially when dealing with large inter-frame motion. Adding the STJS module, we can obtain the PGV with both satisfactory spatial smoothness and temporal consistency (Fig. 11b). We refer readers to the *Appendix* Sec. A.1 for details.

**Scale influence.** Fig. 10a and Fig. 10b show the correspondence ratio decreases with the sequence getting spatially denser and temporally longer, respectively, because the higher spatial and temporal dimensions make the optimization of the spatial and temporal structures more difficult. Note the correspondence ratio is still at a high level.

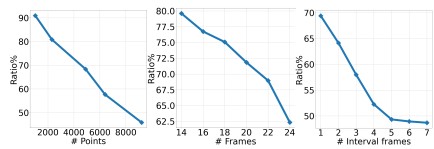

**Robustness to temporal resolution.** We decreased the temporal resolution of original sequences by uniformly selecting one frame every 1 to 7 frames, and the result-

Figure 10: Results of ablation studies on the effect of (**a**) spatial resolution, (**b**) the length of a sequence, and (**c**) the temporal resolution on the performance of our representation framework.

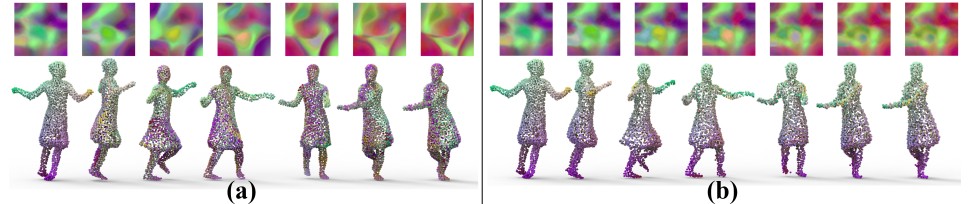

Figure 11: Visual results of ablation studies. (**a**) and (**b**) visualize 7 out of the 20 frames of the resulting PGVs by the FWSS and STJS modules of our framework, respectively. For each group, **Upper**: PGV; **Bottom**: visualization of the PGV in 3D space, where the colors of point clouds indicate the temporal correspondence.

ing sequences contain *larger* non-rigid motion, thus more challenging. As shown in Fig. 10c, our model can maintain a correspondence ratio of higher than 45%, even when the temporal resolution is reduced by seven times.

## 6    DISCUSSION

Although our PGV representation modality has demonstrated impressive performance under various scenarios, there is potential for further enhancement in the following promising aspects:

**Representation quality**. The two primary objectives of our structurization process are intra-frame spatial smoothness and inter-frame temporal consistency. However, as depicted in Figs. 5 and A.9, it is extremely challenging to completely eliminate spatial discontinuity and temporal inconsistency in the generated PGV representation, especially for highly complex shape topologies. It is widely recognized that distortion is inevitable when mapping complex 3D structures to 2D planar domains (Sheffer et al., 2006), as this is fundamentally a parameterization problem. Besides, our current technical pipeline solely relies on the property that convolutional outputs naturally incline to show smooth distributions given smooth inputs. Nevertheless, there is a promising avenue for further improving the representation quality by introducing explicit geometric constraints, such as the local area- or shape-preserving regularization in the spatial domain and the Laplacian smoothing in the temporal domain.

**Computational efficiency**. In our technical implementation, the overall self-reconstructive process is driven by the EMD loss, whose computational complexity grows fast when dealing with the significantly increased number of points. Thus, restricted by the innate inefficient nature of EMD, the scalability of our current framework to large-scale point cloud data is still relatively unsatisfactory. See *Appendix* Sec. A.3 for ablation studies on loss function choices.

**Transformation robustness**. Technically, although our method can straightforwardly achieve translation and scaling invariance via input normalization, it is highly non-trivial to pursue rotation invariance, which itself is a challenging problem (Li et al., 2021a) in the geometry processing community. See *Appendix* Sec. A.3 for some visual results.

**Spatial dimensions of PGVs.** In this paper, we set the spatial dimensions of a PGV to be equal for simplicity, i.e., $H = W$. A more promising solution is adaptively adjusting the spatial dimensions according to the object shape represented by the sequence, which is expected to benefit spatial smoothness.

**Wider applications**. As a regular representation modality that facilitates the adaptation of 2D processing techniques, our PGV opens up many new possibilities for point cloud sequence processing. Notably, geometry compression (Graziosi et al., 2020) might be a promising exploration direction, where we can directly introduce mature image/video codecs. Besides, more low-level (e.g., scene flow estimation (Liu et al., 2019a)) and high-level (e.g., action recognition (Li et al., 2010)) tasks can be incorporated as downstream application scenarios.

## 7    CONCLUSION

We presented a new regular representation modality for 3D point cloud sequences called Point Geometry Video (PGV) for general purposes. PGV is featured with local smoothness in the spatial domain and point-wise correspondence in the temporal domain. We constructed a learning-based framework to conveniently obtain the PGV representation for any sequences in a self-supervised manner. We validated the *effectiveness*, *advantages*, and *universality* of PGV by carrying out various challenging applications. We believe such a representation modality can provide a new structurization strategy for sequences and offer a fresh possibility to overcome the barriers between sequence and 2D video processing and enhance the efficacy of sequence processing pipelines, thanks to its regular spatio-temporal structure.

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

## A  APPENDIX

### A.1  MORE TECHNICAL DETAILS

***Pre-defined 2D and 3D Regular Grids.*** The pre-defined regular 2D and 3D grids are created by regularly sampling coordinates from the first slice and the whole space of $[0, 1]^3$, respectively. For a pre-defined 2D grid $\mathbf{C} \in \mathbb{R}^{H \times W \times 3}$, the values of each pixel coordinate $(u, v)$ are defined as $\mathbf{C}(u, v, 1) = \frac{u-1}{H-1}$, $\mathbf{C}(u, v, 2) = \frac{v-1}{W-1}$, and $\mathbf{C}(u, v, 3) = 0$, where $1 \leq u \leq H$ and $1 \leq v \leq W$. For a pre-defined 3D grid $\mathbf{Q} \in \mathbb{R}^{T \times H \times W \times 3}$, the values of each 3D coordinate $(t, u, v)$ are defined as $\mathbf{Q}(t, u, v, 1) = \frac{t-1}{T-1}$, $\mathbf{Q}(t, u, v, 2) = \frac{u-1}{H-1}$, and $\mathbf{Q}(t, u, v, 3) = \frac{v-1}{W-1}$, where $1 \leq t \leq T$, $1 \leq u \leq H$, and $1 \leq v \leq W$.

***Downstream Task-specific Learning Framework for Sequence Spatial Upsampling.*** Given an input point cloud sequence, we represent it as a PGV. The resulting PGV is further fed into the

subsequent learning framework, as shown in Figure A.1, where frames of the input PGV are spatially upsampled one by one in a residual learning manner. Specifically, for a target PGV frame, we set it as the reference frame and the remaining PGV frames as auxiliary frames. We apply ResNet-style (He et al., 2016) convolution modules with shared weights to extract features from each frame. We also adopt a similar structure to learn the relationship between the features of reference and auxiliary frames and fuse them. Finally, we regress a residual map with the desired higher dimensions from the fused feature maps via the sub-pixel convolution operation, which is further added to the coarsely upsampled frame via bicubic interpolation to produce the upsampled PGV frame.

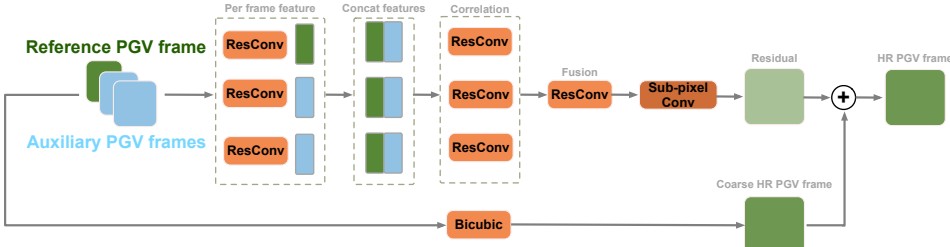

Figure A.1: Flowchart of the framework for sequence spatial upsampling built upon (Jin et al., 2020).

***Downstream Task-specific Learning Framework for Future Point Cloud Frame Synthesis.*** Figure A.2 shows the flowchart of the constructed framework based on SimVP (Gao et al., 2022), where a 2D CNN-based module is used to process each frame separately, and spatio-temporal features are learned through a translator with inception modules (Szegedy et al., 2015). Figure A.3 shows the flowchart of the constructed framework based on FLAVR (Kalluri et al., 2023), where 3D convolution is directly performed on the input PGV to capture spatio-temporal characteristics.

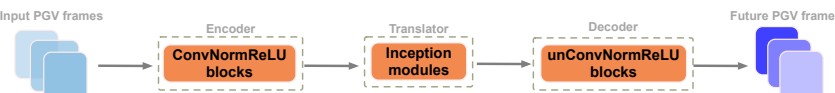

Figure A.2: Flowchart of the framework for synthesizing future point cloud frames of a sequence built upon SimVP (Gao et al., 2022).

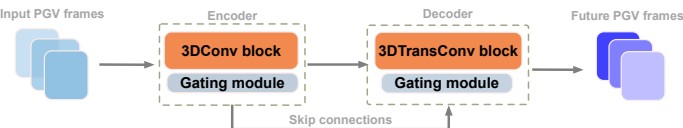

Figure A.3: Flowchart of the framework for synthesizing future point cloud frames of a sequence built upon FLAVR (Kalluri et al., 2023).

***The STJS-only Learning Framework for PGV Generation.*** In Table 2 (the last column) of our manuscript, we demonstrated that directly fitting a sequence from a pre-defined 3D regular grid via Conv3D leads to sub-optimal representation quality. The architecture of the corresponding STJS-only learning framework (i.e., without using the FWSS module) is presented in the following Figure A.4.

## A.2 DETAILED EXPERIMENT SETTINGS

***Correspondence Metric.*** The proposed PGV representation is featured by local smoothness in the spatial domain and point-wise correspondence in the temporal domain. To quantitatively evaluate the dense correspondence promoted by our PGV representation, we correspondingly

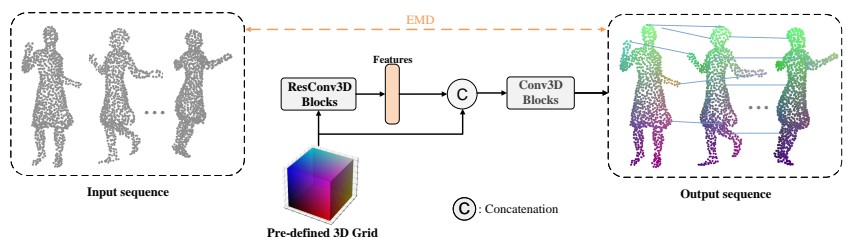

Figure A.4: Flowchart of the framework *without* using the FWSS module.

designed a computational metric to calculate the correspondence ratio, while considering the reconstruction and correspondence accuracy. Recall that our framework represents an *irregular* sequence $\{\mathbf{P}_t \in \mathbb{R}^{N \times 3}\}_{t=0}^{T-1}$, where point-wise correspondence across frames is *unknown*, as a PGV with $T$ frames, denoted as $\{\mathbf{G}_t \in \mathbb{R}^{H \times W \times 3}\}_{t=0}^{T-1}$ with $H$ and $W$ being the height and width of the 2D color video frame and $H = W = \sqrt{N}$. We proposed Algorithm 1 to calculate the ratio of semantically corresponded points of the obtained PGVs.

---

**Algorithm 1** Algorithm for calculating correspondence ratio

---

**Require:** $\mathcal{G}_t, \mathcal{P}_t, T, N, k$

  $t \leftarrow 0$

  **for** $t < T$ **do**                      ▷ loop for checking sequence reconstruction accuracy

      $\mathcal{G}_t \leftarrow \text{index}(\mathcal{P}_t, \text{knn}(\mathcal{P}_t, \mathcal{G}_t, 1))$  ▷ re-matching the original points for checking the per-frame reconstruction accuracy

      $t += 1$

  **end for**

  **function** $\text{R}(a, b, k)$                               ▷ pairwise ratio function

      $v \leftarrow 0$

      $\text{cnt} \leftarrow N$

      **while** $\text{cnt} \neq 0$ **do**

         $s_a = \text{knn}(a, a, k)$                   ▷ graph on source point cloud

         $s_b = \text{knn}(b, b, k)$                   ▷ graph on query point cloud

         $v += \frac{\text{len}(s_a \& s_b)}{k}$          ▷ correspondence ratio in a local region

         $\text{cnt} \leftarrow \text{cnt} - 1$

      **end while**

      $v \mathrel{/}= N$

      **return** $v$

  **end function**

  $\text{ratio} \leftarrow 0$

  $t \leftarrow 0$

  **for** $t < T$ **do**                     ▷ loop for checking sequence correspondence accuracy

      $\text{ratio} += \frac{1}{2}(R(\mathcal{G}_t, \mathcal{G}_{t-1}, k) + R(\mathcal{G}_{t-1}, \mathcal{G}_t, k))$ ▷ forward and backward correspondence ratio for adjacent frames

      $t += 1$

  **end for**

  **return** $\text{ratio} \mathrel{/}= T$

---

***Adaptation of Neuromorph (Eisenberger et al., 2021) and Corrnet3D (Zeng et al., 2021) for Sequence Correspondence Prediction.*** As involved in Section 5.1 of our manuscript, the compared methods Neuromorph (Eisenberger et al., 2021) and Corrnet3D (Zeng et al., 2021) were primarily designed for processing a pair of 3D shapes. We made necessary adaptations to the two methods to enable the prediction of correspondence across sequence frames. For Neuromorph (Eisenberger et al., 2021), we set the number of interpolated in-between frames to 1, so the correspondence from

the interpolated frame to the neighboring frame can be built in each step. Then, we slide the model frame-by-frame to obtain the sequence correspondence. Similarly, for Corrnet3D (Zeng et al., 2021), we start from the first frame to learn the correspondence between the first and second frames and then slide frame-by-frame to achieve the sequence correspondence.

***Benchmark Datasets.*** Here we provide more details about the sequences used in our experiments. As shown in Table A.1, the first ten sequences of the left side are from the MIT-Animation (Vlasic et al., 2008) dataset, with the number of frames varying from 150 to 250, and the last six sequences are from 8iVSLF (Krivokuca et al., 2018), with the number of frames as 300. Moreover, the first 13 sequences of the right side are the non-rigid animal poses and facial expressions from DTTM (Sumner & Popović, 2004) with the number of frames varying from 9 to 53, and the remaining sequences are from Owlii (Xu et al., 2017) with the number of frames as 600.

Table A.1: Details of the dataset from MIT-Animation (Vlasic et al., 2008), 8iVSLF (Krivokuca et al., 2018), DTTM (Sumner & Popović, 2004) and Owlii (Xu et al., 2017) used in our experiment.

| MIT-Animation & 8iVSLF | # Frames | DTTM & Owlii | # Frames |
|---|---|---|---|
| bouncing | 175 | camel-collapse | 53 |
| crane | 175 | camel-gallop | 48 |
| handstand | 175 | camel-poses | 10 |
| jumping | 150 | cat-poses | 9 |
| march 1 | 250 | elephant-gallop | 48 |
| march 2 | 250 | elephant-poses | 10 |
| samba | 175 | face-poses | 9 |
| squat 1 | 250 | flamingo-poses | 10 |
| squat 2 | 250 | head-poses | 9 |
| swing | 150 | horse-collapse | 53 |
| longdress | 300 | horse-gallop | 48 |
| loot | 300 | horse-poses | 10 |
| redandblack | 300 | lion-poses | 9 |
| soldier | 300 | basketball player | 600 |
| thaidancer | 300 | dancer | 600 |
| boxer | 300 | exercise | 600 |
| - | - | model | 600 |

***Downstream task Evaluation of Sequence Spatial Upsampling.*** In this experiment, the sparse input sequences and ground-truth dense sequences contain 625 and 2500 points in each point frame, respectively. The compared methods were supervised by the per-frame Earth Mover's Distance (EMD) loss implemented in (daerduoCarey, 2022), and our method by $L_1$ loss owing to the regularity of PGVs. We used the metric of averaged EMD and mNUC per frame to verify the reconstruction accuracy and uniformity of the output sequence, respectively.

***Downstream Task Evaluation of Future Point Cloud Frame Synthesis.*** In this experiment, we used the 1024-point, irregular sequences sampled from the original sequence with FPS (Eldar et al., 1997) during the training phase. Eight consecutive frames were supplied, with the first five serving as input frames and the last three serving as ground truth frames. To assess the effectiveness of each method, we compared the predicted sequence with the actual sequence during the testing period. To check the reconstruction's correctness and the output sequence's uniformity, we employed the EMD and mNUC metrics averaged per frame.

## A.3  ADDITIONAL VERIFICATIONS AND DISCUSSIONS

***Loss Function.*** In Section 6 of the manuscript, we discussed the computational efficiency of our framework, which is driven by the EMD loss. In fact, another alternative choice is the CD loss. Thus, to quantitatively confirm the superiority of the EMD loss in the PGV framework, we performed an ablation study by analyzing the impact of loss functions on the overall PGV framework. As shown

in Table A.2, switching to the CD loss as supervision causes significant performance degradation. The EMD loss turns to be much more effective in terms of promoting the temporal consistency of PGVs.

Table A.2: Effect of loss functions on the whole PGV framework. The table presents the performance of the PGV framework in terms of correspondence ratio (%).

| Loss Functions | Correspondence Ratio (%) |
|---|---|
| EMD | **92.06** |
| CD | 47.04 |

***Multi-scale fitting manner***. To confirm the effectiveness of the multi-scale fitting approach in the FWSS module of the PGV framework, we conducted an ablation study by analyzing the impact of fitting scales on the FWSS module. Specifically, we represented a point cloud frame of the *Swing* sequence as a 2D color image and reported the EMD, CD, and spatial smoothness ratio (SSR) (%).[4] As indicated in Figure A.5, multi-scale EMD achieves better spatial smoothness in the resulting 2D color image.

***Potential Solutions to Overcome Computational bottleneck***. As mentioned in Section 6 of the manuscript, improving computational efficiency is challenging due to the complex EMD loss used in our framework, limiting our scalability to large-scale point cloud data. One possible solution is to use the grid-based loss (Xie et al., 2020) to accelerate the learning process. However, voxel resolutions still limit representation power and lead to redundancy for surface shapes. Further research is needed to develop more efficient loss functions that can handle large-scale point cloud data without sacrificing representation quality.

***Transformation Sensitivity***. Achieving rotation invariance is a longstanding and highly non-trivial problem (Zhang et al., 2019; 2020; Xu et al., 2021a) in the geometry processing community. And our current technical implementation of the whole PGV generation framework also suffers from sensitivity to transformation. As shown in Figure A.6, we provide the visualization of the 2D color images from the FWSS module by rotating the input point cloud frame randomly, showing that the patterns of the resulting 2D color images differ significantly under different rotations.

---

[4]We begin by selecting a random pixel within the 2D color image and define a patch around it. Next, we re-organize the image in 3D space by selecting the corresponding 3D point and taking a neighborhood around it. We then compare the ratio of overlapping points between the patch and the 3D neighborhood.

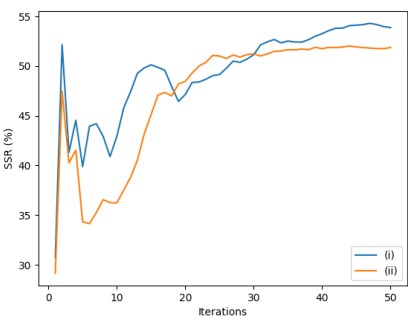

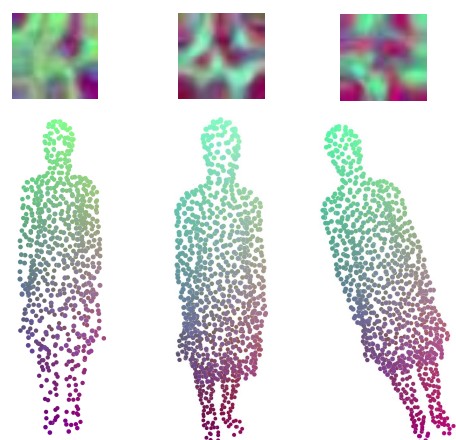

Figure A.5: Comparison of fitting the FWSS module under different scales using various metrics. The two strategies evaluated are (**i**) multi-scaled EMD loss and (**ii**) single-scaled EMD loss. Multi-scale refers to the FWSS module driven by EMDs at three different scales as shown in Fig. 3 of the manuscript, while single-scale only uses the largest scale.

Figure A.6: Visual results of the 2D color image produced by the FWSS module applied to rotated input point clouds with 3 random angles.

## A.4 MORE VISUAL RESULTS

We provide more visualizations on the DTTM (Sumner & Popović, 2004) dataset in Figures A.7 and A.8, where it can be seen that, despite the large non-rigid motion deformation, the obtained PGVs still demonstrate dense correspondence across frames very well. Notably, the results highlight the framework's aptitude in effectively processing scanned LiDAR sequences, as evidenced by its ability to establish dense correspondence across frames.

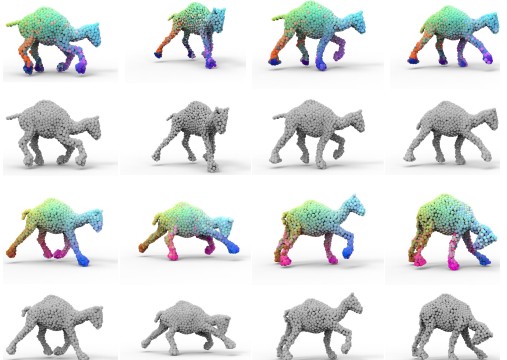

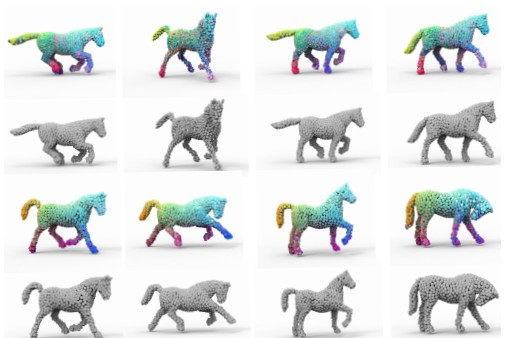

Figure A.7: More visualizations on the sequence *camel-poses*. The **odd** rows visualize the resulting PGV in 3D space, where the colors indicate dense correspondence across frames. The **even** rows are the corresponding input sequences with **unknown** dense correspondence across frames.

Figure A.8: More visualizations on the sequence *horse-poses*. The **odd** rows visualize the resulting PGV in 3D space, where the colors indicate dense correspondence across frames. The **even** rows are the corresponding input sequences with **unknown** dense correspondence across frames.

We provide more visualization of the resulting PGV for a sequences of rotating *torus* (Figure A.10) and *multiple objects* (Figure A.11). These examples feature either holes or multiple objects. In Figure A.10, it is evident that the generated PGV sequence maintains spatial smoothness within local areas of the PGV frames and temporal consistency among different frames of point clouds, as indicated by the colors on the points for the challenging shape *torus*. In Figure A.11, for the generated PGV of the sequence with multiple objects, it is observable that each object is automatically mapped

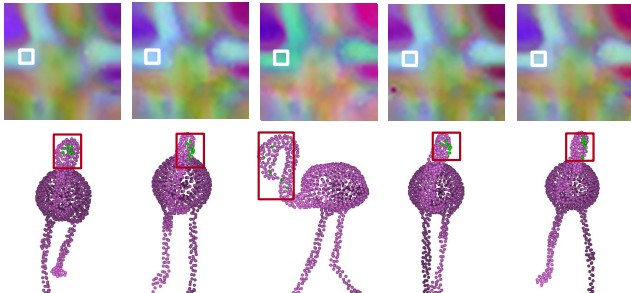

Figure A.9: Visual results of the sequence *Flamingo*. **Upper**: PGV. **Bottom**: visualization of the above PGV in 3D space, where the points in green correspond to the PGV region in the white boxes.

to a patch region on the PGV frames. Moreover, the correspondence of any one object across different frames of the point clouds is still preserved, as demonstrated by the colors on the points of different objects.

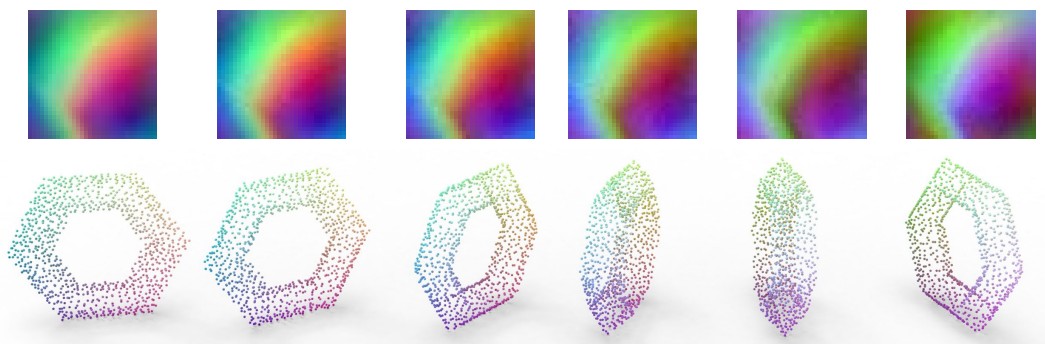

Figure A.10: Visual results of the PGV generated by our framework on a sequence of the rotating *torus*. **Upper**: PGV; **Bottom**: visualization of the PGV in 3D space, where the colors of point clouds indicate the temporal correspondence.

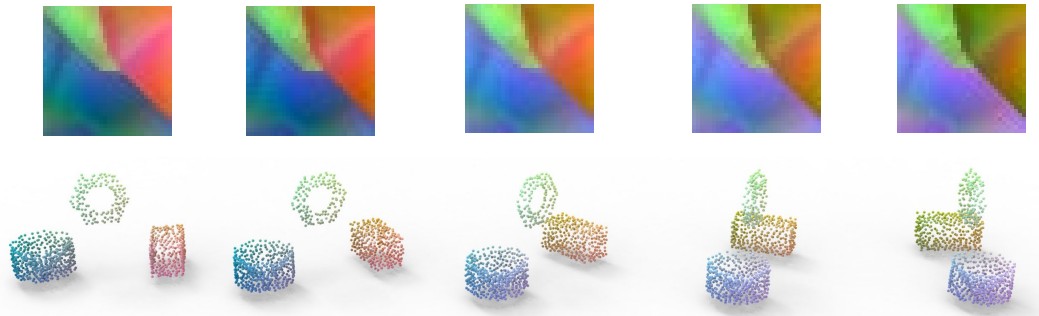

Figure A.11: Visual results of the PGV generated by our framework on a sequence of the rotating *multiple objects*. **Upper**: PGV; **Bottom**: visualization of the PGV in 3D space, where the colors of point clouds indicate the temporal correspondence.

We also present visual results of the PGV generated by our framework on *Swing* to demonstrate the representation quality for both partial point clouds and long sequences. We fixed the camera perspective and extracted visible points from the *Swing* sequence to form partial point clouds, subsequently represented as PGV. We visualized 20 frames exhibiting significant motion from this sequence, displayed in a row scanning order.

Although the representation quality of PGVs may degrade to some extent, due to the fact that incomplete/partial sequences inherently pose much greater representation difficulties, there are no essential barriers for our technical scheme to process such challenging data. The additional experimental re-

sults in Fig. A.12 of the updated pdf file demonstrate that our method works very well on partial point cloud data.

We proposed a new PGV representation structure together with a specialized learning framework to implement such a point cloud sequence structurization process. Pursuing long-term performance tends to be an engineering issue, just like the standard "Low-Delay-P (IPPP)" working mode in video compression, where a video sequence is divided into groups of pictures (GOPs).

In our method, for instance, we might set up a fitting window that spans five frames, designed to overfit from frame $i$ to $i + 4$. Then, this window shifts to start at frame $i + 4$, handling frames $i + 4$ to $i + 8$, and so forth. This approach allows us to methodically fit the entire long sequence in a chunk-by-chunk fashion, ensuring intra-frame smoothness and inter-frame temporal consistency within each chunk. For the transitions between chunks, owing to our overfitting mechanism's high level of reconstructive fidelity, we align adjacent chunks by applying a nearest-neighbor strategy at the overlapping frames of consecutive chunks. Ultimately, this yields a complete long sequence where each frame is characterized by both spatial smoothness and temporal consistency.

Specifically, as illustrated in Fig. A.12, the PGV generated from partial point cloud sequences demonstrate local smoothness in each frame and notable temporal consistency, despite the primary variation due to missing points subtly altering color block distributions. For instance, the absence of green points, as marked by the red box, results in a reduction of green within the PGV frames. These areas are smoothly replaced by the colors of nearby points. Apart from the boundaries of missing points, color shifts due to absent points in the local area of the PGV frame transition smoothly rather than changing abruptly. Additionally, even in longer sequences, the PGV frames exhibit a largely consistent pattern, with variations occurring mainly in areas of substantial motion like the hands. The uniformity in color distribution within each frame further underscores the spatial smoothness of the PGV representation. This consistency underscores the effectiveness of our approach in maintaining spatial and temporal smoothness, even in dynamically moving sequences. Such spatio-temporal regularization facilitates efficient indexing for convolution operations in downstream tasks, such as sliding window techniques, and aids in feature learning.

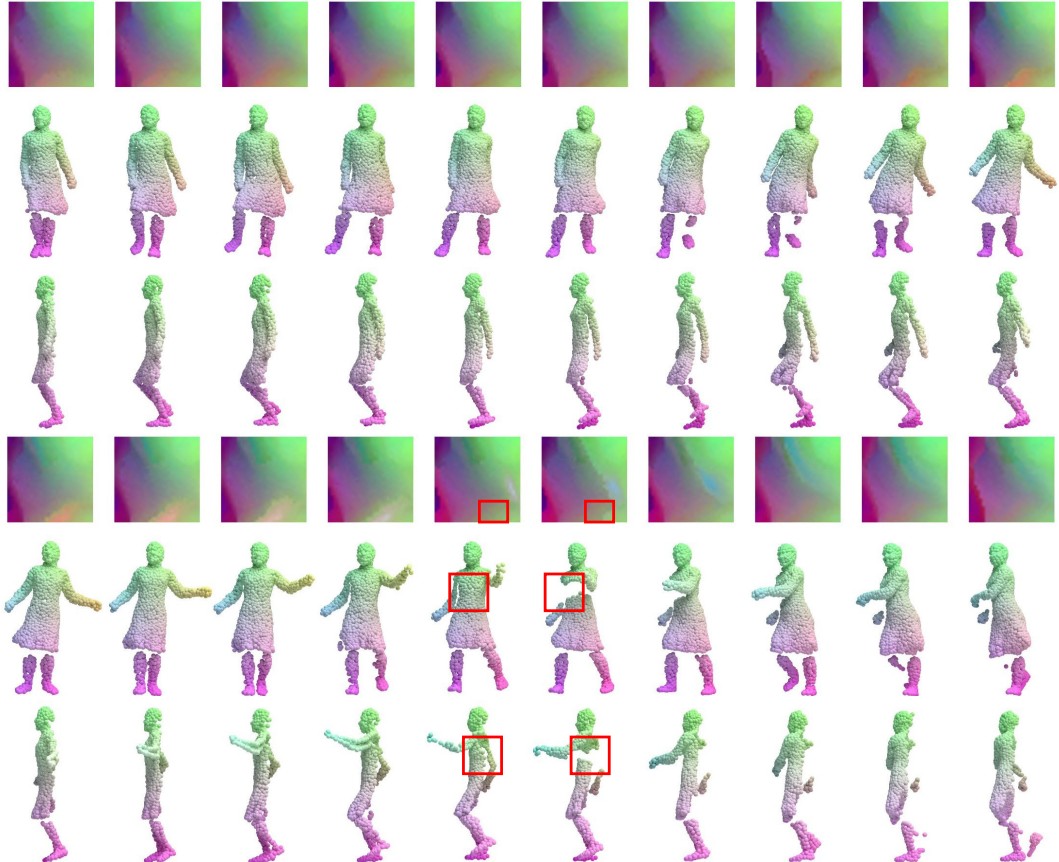

Figure A.12: Visual results of the PGV generated by our framework on a long sequence of partial point clouds from *Swing*, displayed in row-scanning order. The **first** and **fourth** rows showcase the generated PGV. The other rows display the PGV in 3D space, where the colors of the point clouds indicate temporal correspondence, with both the front (**second** and **fifth** rows) and side views (**third** and **sixth** rows).

