# OpenReview forum: "3D Point Cloud Sequences as 2D Videos"
_ICLR.cc/2024/Conference — Submitted to ICLR 2024_

### Official Review · Reviewer_3RY6 · 2023-10-26

**Soundness:** 3 good
**Presentation:** 3 good
**Contribution:** 2 fair
**Rating:** 8
**Confidence:** 5

**Summary:**

This work proposes a new way to represent 3D point clouds called point geometry video (PGV). It is to "flatten" the original 3D point cloud onto a 2D image plane, on which the [R, G, B] channels of each pixel are used to carry the [x, y, z] coordinate of the associated point in 3D space. The author imposes spatial and temporal smoothness in the PGV representation via an overfitting scheme for training the neural network modules. The resulting PGV representation of the 3D point cloud sequence is applied in dense correspondence prediction, spatial up-sampling, and future point cloud frame synthesis, showcasing interesting results.

**Strengths:**

This proposed PGV representation of point cloud sequence appears to be interesting, it "collapses" the 3D point clouds to regular 2D images so that they can be consumed by a regular processing pipeline (although I have concerns about its novelty, see weakness). The paper is well-written overall and easy to follow.

**Weaknesses:**

Despite the strength of the work, there are quite a few problems with it:

1. The concept of this "point geometry video" is not entirely new. In fact, it is very similar to two international patents published in 2022:

(a). "Learning-based point cloud compression via tearing transform," with International Publication Number: WO2022232547A1 (available via https://patentimages.storage.googleapis.com/51/19/af/ee3e55ed75618b/WO2022232547A1.pdf),

(b). "Learning-based point cloud compression via unfolding of 3D point clouds" with International Publication Number: WO2022271602A1 (available via https://patentimages.storage.googleapis.com/71/49/ab/6e2ea50f6af4a1/WO2022271602A1.pdf).

Idea-wise, this PGV work is especially similar to (b), the "unfolding" work. Because the "unfolding" work exactly proposes to view the 3D coordinates of the point clouds as pixel values on images, so that they can be processed by downstream 2D pipelines, e.g., see Fig. 3 of the "unfolding" work.
Additionally, the core idea to create such images in the PGV work (Fig. 3 in PGV paper) and that of the "unfolding" work is basically the same (Fig. 3 in "unfolding" work) - both intend to "deform" a predefined regular 2D grid to match with the original point cloud.
I can identify that the detailed designs of the two works are different (e.g., detailed architecture difference, overfitting in PGV v.s. latent codeword for representation in "unfolding" work), and the PGV work as an additional step to promote temporal consistency (Fig. 4 of the PGV paper). However, the idea of the PGV proposal still has many overlaps with that of the "unfolding" work, weakening the novelty of the PGV work.

2. The method used in Fig. 3 (FWSS) is highly related to a thread of point cloud decoders based on deforming a pre-defined pattern like a 2D grid. Representative works in this thread include AtlasNet, FoldingNet, PointCapsuleNet, etc., may be informative to also discuss these works.

3. The overfitting scheme in this work can limit the practical usage of the proposal due to high computational complexity, i.e., the network needs to be retrained for every new point cloud sequence.

4. How does the proposed method behave if the point clouds to be processed contain more complicated topologies? E.g., more holes, or multiple objects. The 8i sequences being used in the experimentation may be somewhat limited due to their simplicity.

**Questions:**

Please refer to the weaknesses. Please especially address my concern regarding the novelty of the proposal.

---

> ### Author Response · Authors · 2023-11-18
> **Response to Reviewer 3RY6 (1/2)**
>
> **Response to Reviewer 3RY6 (1/2)**
>
> Thank you for your time and effort in reviewing our manuscript and providing constructive comments. In the following, we address your concerns comprehensively.
>
> > ### Comment 1: The major concern I have is regarding its novelty - the concept of this "point geometry video" is not entirely new. In fact, it is very similar to two international patents published in 2022. Idea-wise, ...
>
> **Response**: Thanks for pointing out the two patents we were not previously aware of. However, after carefully examining the links provided by the reviewer, we found that both patents propose the method for single point cloud compression, which is not similar to our goal, i.e., structuring the irregular 3D point cloud sequences into regular point geometry video.
> Besides, the concept of representing a 3D geometric model with a 2D RGB image was first demonstrated by the paper 'Geometry Images' in 2002.
> The challenge lies in designing effective deep learning methods to implement this unfolding process, which varies across different papers. It is crucial to emphasize that our method diverges from this concept by not focusing on individual point clouds. Instead, it processes entire 3D point cloud sequences, aiming to maintain both intra-frame smoothness and inter-frame temporal consistency. The **most significant novelty** of our PGV representation lies in being the first to structure irregular 3D point cloud sequences using only their coordinates.
>
> > ### Comment 2: The method used in Fig. 3 (FWSS) is highly related to a thread of point cloud decoders based on deforming a pre-defined pattern like a 2D grid. Representative works in this thread include AtlasNet, FoldingNet, PointCapsuleNet, etc., may be informative to also discuss these works.
>
> **Response**: First of all, the FWSS module is **technically different** from the mentioned works, AtlasNet, FoldingNet, and PointCapsuleNet, because they encode the input point cloud into a latent codeword, while FWSS directly takes the pre-defined grid and overfits the ground truth point cloud. We acknowledge that the reviewer has already identified this, as mentioned in the comments 'overfitting in PGV vs. latent codeword'. Besides, although the mentioned works can achieve a certain degree of generalization ability, they still suffer from relatively low fidelity between the generated point clouds and the ground truth ones. In our PGV representation, we choose the overfitting manner implemented with a multi-scale structure because the PGV demands **high reconstruction fidelity** to 're-organize' or 'sort' the ground truth point cloud to generate its structured format.
> We have updated the related work section of the manuscript to discuss these works. However, these approaches are confined to **static** geometric representations and thus **unsuitable** for dynamic point cloud sequences, which require spatio-temporal structurization. To the best of our knowledge, our representation is the **first attempt** at structuralizing dynamic point cloud sequences. Transitioning from static point clouds to dynamic sequences introduces additional challenges due to the unstructured nature in both spatial and temporal domains. We address these challenges by proposing the PGV representation, which fills the gaps involved in cross-modal learning between 3D point cloud sequences and 2D videos.
>
> Furthermore, the FWSS module alone does not perform well. As shown in Table 2 of our paper, the absence of the STJS module, i.e., when relying solely on FWSS, significantly degrades the learned correspondence. As visualized in Fig. 11, while the FWSS module can generate 2D color images with spatial smoothness, it lacks temporal consistency (see Fig. 11a), especially when dealing with large inter-frame motion. However, by adding the STJS module, we achieve PGV with both satisfactory spatial smoothness and temporal consistency (see Fig. 11b). More details are provided in Appendix Section A.1.

---

> > ### Comment · Reviewer_3RY6 · 2023-11-19
> > **Regarding novelty of the proposal**
> >
> > I appreciate the author's response to my concerns about the novelty of PGV. In a word, I am basically convinced that the novelty of this work is sufficient for acceptance in ICLR.
> >
> > On one hand, this work indeed puts more emphasis on handling dynamic point cloud sequences which are non-trivial and differ from the works I have listed. On the other hand, the FWSS differs from AtlasNet, FoldingNet, etc., in the sense that it overfits the point cloud sequences to achieve better reconstruction quality. Additionally, the STJS module which enforces temporal consistency across video frames is also a novel component that does not exist in former works.
> >
> > Aside from the "Geometry Images", the AtlasNet, FoldingNet, and the two patents I listed, I realized the PGV paper is also related to this work: "Folding-based Compression of Point Cloud Attribute" by M. Quach et al. which uses the folding/deformation-based technique to map the attributes on to an 2D image. I encourage the author to also include this work when discussing related research. Additionally, it will also be helpful to briefly mention the relationship of this work to V-PCC in point cloud compression.
> >
> > Note that I have updated my rating of the paper accordingly.

---

> ### Author Response · Authors · 2023-11-18
> **Response to Reviewer 3RY6 (2/2)**
>
> **Response to Reviewer 3RY6 (2/2)**
>
> > ### Comment 3: The overfitting scheme in this work can limit the practical usage of the proposal due to high computational complexity, i.e., the network needs to be retrained for every new point cloud sequence.
>
> **Response**:
> We have mentioned the issue of computational complexity in the discussion section of our paper (Section 6). Although our method is still limited by the efficiency of the EMD loss, we believe there is potential for improvement with the advancement of loss functions. Taking the popular NeRF method as an example, it initially faced high computational complexity upon its proposal, but subsequent NeRF-based works have improved its efficiency.
>
> From a different perspective, successfully generating PGV itself implies time savings, as the spatio-temporally regularized PGV representation described in our paper lacks a real-world ground truth and is even more challenging to achieve through human annotation. In other words, there are no readily available ground truth sequences that exhibit both spatial smoothness and temporal consistency. Even if it were possible to annotate them, creating such sequences through human annotation would require tedious efforts and consume a significant amount of time.
>  Additionally, the PGV representation can be used in an **offline** manner to **automatically** process irregular 3D point cloud sequences, and the resulting PGVs will be **permanently** stored, on which the conducted downstream tasks can greatly benefit in terms of efficiency, performance, and implementation.
>
> Last but not least, being an early attempt to apply spatio-temporal regularization to 3D point cloud sequences, we argue that solving the fundamental challenge of creating these regularized sequences from scratch is more crucial than addressing efficiency issues.
>
> > ### Comment 4: How does the proposed method behave if the point clouds to be processed contain more complicated topologies? E.g., more holes, or multiple objects. The 8i sequences being used in the experimentation may be somewhat limited due to their simplicity.
>
> **Response**:  We add the resulting PGV for a sequences of rotating *torus* (Figure A.10) and *multiple objects* (Figure A.11) in our uploaded revised manuscript. These examples feature either holes or multiple objects.
> In Fig. A.10, it is evident that the generated PGV sequence maintains spatial smoothness within local areas of the PGV frames and temporal consistency among different frames of point clouds, as indicated by the colors on the points for the challenging shape *torus*.
> In Fig. A.11, for the generated PGV of the sequence with multiple objects, it is observable that each object is automatically mapped to a patch region on the PGV frames. Moreover, the correspondence of any one object across different frames of the point clouds is still preserved, as demonstrated by the colors on the points of different objects.
>
> As we discuss in **Section 6** of our manuscript regarding representation quality, completely eliminating spatial discontinuity and temporal inconsistency in the generated PGV, especially for highly complex shape topologies, is an extremely challenging task due to its non-trivial nature in parameterization. We believe this can be improved by introducing explicit geometric constraints, such as local area- or shape-preserving regularization in the spatial domain and Laplacian smoothing in the temporal domain. In our experiments, we observed that even with non-perfect smoothness and correspondence, the PGV representation still significantly enhances performance in downstream tasks (refer to results in **Sections 5.2** and **5.3**).

---

> > ### Comment · Reviewer_3RY6 · 2023-11-19
> > **Regarding computational cost and applicability**
> >
> > Thank you for the comprehensive answers to my concerns. I agree that the complexity of PGV can be potentially improved, like in the cases of NeRF-based networks. It also has good potential to benefit downstream tasks due to the regularity in the 2D video frames. Additionally, the informative example of "torus" and "multiple objects" also alleviates my concerns when PGV is applied to complicated scenes. Again, I do see the good potential of applying this representation to different point cloud processing tasks.

---

> > > ### Author Response · Authors · 2023-11-20
> > >
> > > We sincerely thank the reviewer for taking the time and effort to review our paper. We deeply appreciate your recognition of our efforts, constructive comments, and recommendations. Your confirmation of the potential of our method greatly encourages us. Thank you.

---

### Official Review · Reviewer_BEST · 2023-10-27

**Soundness:** 3 good
**Presentation:** 3 good
**Contribution:** 4 excellent
**Rating:** 6
**Confidence:** 4

**Summary:**

This paper proposes a new regular representation modality for 3D point cloud sequences, namely Point Geometry Video (PGV). Specifically, the frame-wise spatial structurization (FWSS) module learns to map a 3D point in the point cloud to a 2D regular grid with 2D convolutions in a coarse-to-fine manner, and the spatio-temporal joint structurization (STJS) module learns to add temporal consistency to the PGV with 3D convolutions. Extensive experiments are conducted on several downstream tasks, namely, dense correspondence prediction, spatial up-sampling, and future point cloud frame synthesis, showing good qualitative and quantitative results.

**Strengths:**

1. The proposed method is novel and insightful. The intuition is similar to UV mapping in a learnable way.
2. The qualitative and quantitative results are good. The method is promising to facilitate downstream tasks.

**Weaknesses:**

1. I would recommend the authors put the symbols with their corresponding items on the figures. For instance, in section 3.2 we see the $\mathbf{S}^0$ is the pre-defined 3D grid corresponding to the cube in Fig. 4. But what are $\mathbf{S}_f$, $\mathbf{S}_d$, and $\mathbf{S}'$ referring to? Readers would be happy to see them clearly noted in the figure.
2. For me, the convolutional manner in the STJS module might not be proper. Since the FWSS results are without temporal consistencies, it's surprising that local convolution kernels are able to handle irrelevant information across different timesteps.

**Questions:**

1. Can the authors explain in detail how the flow maps in the STJS module are learned?
2. Can the method deal with partial point clouds, incomplete data, or irregular data? Say if we have a point cloud sequence unprojected from a video, we may want to take as inputs partial point clouds due to occlusions, and some parts in one frame might have no correspondence in some other frames.

---

> ### Author Response · Authors · 2023-11-18
> **Response to Reviewer BEST (1/2)**
>
> **Response to Reviewer BEST (1/2)**
>
> Thank you for your time and effort in reviewing our manuscript and providing constructive comments. In the following, we will comprehensively address your concerns
>
> > ### Comment 1: I would recommend the authors put the symbols with their corresponding items on the figures. For instance, in section 3.2 we see the $\mathbf{S}^0$ is the pre-defined 3D grid corresponding to the cube in Fig. 4. But what are $\mathbf{S}_f$, $\mathbf{S}_d$, and $\mathbf{S}'$ referring to? Readers would be happy to see them clearly noted in the figure.
>
> **Response**:
> Thank you for your recommendation.
> In our figures, $\mathbf{S}_f$ represents the learned 'Flow Maps' resulting from the 'Conv3D Blocks', as depicted in Fig. 4. $\mathbf{S}_d$ denotes the stack of copies of the first frame $\mathbf{I}_0$ emanating from the FWSS module, positioned to the right of the FWSS module and the pre-defined 3D Grid in Fig. 4. Meanwhile, $\mathbf{S}'$ corresponds to the stack of the remaining frames of the final PGV, located in the black box beneath the summation (`+') operation in Fig. 4.
> We acknowledge the importance of clarity in our figures and will incorporate these symbols in Fig. 4 in our revised draft.
>
> > ### Comment 2: For me, the convolutional manner in the STJS module might not be proper. Since the FWSS results are without temporal consistencies, it's surprising that local convolution kernels are able to handle irrelevant information across different timesteps.
>
> **Response**:
> We have investigated the effects of FWSS and STJS by visualizing their outputs, as depicted in Fig. 11 of the paper. It can be seen that the coarse PGV, produced solely by FWSS, already shows similar patterns in consecutive frames, indicating the presence of coarse temporal consistency along the time dimension. Although this level of temporal consistency may not be as precise as that achieved with the STJS module, the intra-frame spatial smoothness and the inter-frame temporal consistency within the small time windows can aid in spatial-temporal feature extraction, particularly when the Conv3D blocks slide along both the spatial and temporal dimensions.

---

> ### Author Response · Authors · 2023-11-18
> **Response to Reviewer BEST (2/2)**
>
> **Response to Reviewer BEST (2/2)**
>
> > ### Comment 3: Can the authors explain in detail how the flow maps in the STJS module are learned?
>
> **Response**:
> As illustrated in Fig. 4, the flow maps in the STJS module are generated by a decoder composed of Conv3D blocks, denoted as $h\_{{\phi}}\^d(\cdot)$.
> This decoder deforms a predefined 3D grid, $\mathbf{S}\^0$, which is concatenated with the encoded residual features from the ResConv3DBlocks encoder, $h\_{{\phi}}\^e(\cdot)$.
> The three-layer residual-style encoder processes the remaining frames from the FWSS module and embeds both intra- and inter-frame information into the feature embedding. This is subsequently followed by a decoder with three layers of Conv3D blocks to generate the flow maps.
> As detailed in Section 4.2, this process can be formulated as $\mathbf{S}\_f = h\_{{\phi}}\^d(\texttt{CAT}(h_{{\phi}}\^e(\mathbf{S}_r),\mathbf{S}\^0))$, where $\texttt{CAT}(\cdot)$ denotes the concatenation operation, and $\mathbf{S}_r$ represents the remaining frames from the FWSS module.
>
> > ### Comment 4: Can the method deal with partial point clouds, incomplete data, or irregular data? Say if we have a point cloud sequence unprojected from a video, we may want to take as inputs partial point clouds due to occlusions, and some parts in one frame might have no correspondence in some other frames.
>
> **Response**: Converting irregular 3D point cloud sequences into regular ones entails an approximation trade-off. As the number of points and frames increases, enhancing spatial and temporal information respectively, our PGV more closely approximates regular data with accurate correspondence. This increased precision results from more sampling points along the time dimension, enabling the network to align better with the actual trajectories. However, fewer samples in partial and incomplete data lead to challenges in achieving temporal consistency close to real trajectories. Despite this, our method still facilitates a structured format to approximate correspondences.
>
> The mixed sequence in Fig. 5(a) (bottom) and the Lidar sequence of Fig. 8 demonstrate our method's performance in irregular scenarios.
> In Fig. 5(a) (bottom), while the basketball in the second frame is absent in others, the basketball and hand areas across different frames display a form of correspondence.
> Such correspondence, though not exactly semantically correct, can still serve to enhance the structuredness of the resulting sequences.
> Fig. 8 showcases a scenario where a car-mounted Lidar scans point clouds while moving along a street. The resulting PGV (Fig. 8(b)) and the point cloud sequence format of the PGV (Fig. 8(c)) reveal that, although some points are missing as frames evolve, the most related parts of the point cloud still correspond, as indicated by similar patterns in the PGV.
>
> PGV offers a compact, structured format for partial and incomplete sequences, enhancing feature extraction in downstream tasks. Nonetheless, the accuracy of correspondence improves with more complete and coherent input sequences, as evidenced by the ablation study in Figure 10.

---

> ### Comment · Reviewer_BEST · 2023-11-20
>
> Thank you for your detailed response and corresponding modifications to the draft. They somewhat make sense to me. I still have concerns about the long-term performance and the ability to handle incomplete/partial data of this method. Thus I will keep my original ratings given its potential applications and the limitations the authors agreed on.

---

> > ### Author Response · Authors · 2023-11-21
> >
> > Thanks for your time in reading our responses and active communication. We appreciate your recognition of our effort and the favorable recommendation.
> >
> > As for the two concerns you particularly raised, i.e., (1) the long-term performance, and (2) the ability to handle incomplete/partial data, below we will additionally supplement the targeted experimental results and provide more clarifications and explanations to further facilitate your assessment of our approach.
> >
> > Foremost, although the representation quality of PGVs may degrade to some extent, due to the fact that incomplete/partial sequences inherently pose much greater representation difficulties, there are **no essential barriers** for our technical scheme to process such challenging data.
> > **More importantly**, the additional experimental results in Fig. A.12 of the updated pdf file demonstrate that our method works very well on partial point cloud data.
> >
> > Second, from our point of view, the most valuable contribution of this work is to propose a new PGV representation structure together with a specialized learning framework to implement such a point cloud sequence structurization process. Pursuing long-term performance tends to be an engineering issue, just like the standard ``Low-Delay-P (IPPP)'' working mode in video compression [R1], where a video sequence is divided into groups of pictures (GOPs).
> >
> > -- [R1] B. Bross, et al., "Overview of the Versatile Video Coding (VVC) Standard and its Applications," in IEEE TCSVT, 2021.
> >
> > In our method, for instance, we might set up a fitting window that spans five frames, designed to overfit from frame $i$ to $i+4$. Then, this window shifts to start at frame $i+4$, handling frames $i+4$ to $i+8$, and so forth. This approach allows us to methodically fit the entire long sequence in a chunk-by-chunk fashion, ensuring intra-frame smoothness and inter-frame temporal consistency within each chunk.
> > For the transitions between chunks, owing to our overfitting mechanism's high level of reconstructive fidelity, we align adjacent chunks by applying a nearest-neighbor strategy at the overlapping frames of consecutive chunks. Ultimately, this yields a complete long sequence where each frame is characterized by both spatial smoothness and temporal consistency.
> >
> > Specifically, as illustrated in Fig. A.12, the PGV generated from partial point cloud sequences demonstrates local smoothness in each frame and notable temporal consistency, despite the primary variation due to missing points subtly altering color block distributions. For instance, the absence of green points, as marked by the red box, results in a reduction of green within the PGV frames. These areas are smoothly replaced by the colors of nearby points.
> > Apart from the boundaries of missing points, color shifts due to absent points in the local area of the PGV frame transition smoothly rather than changing abruptly.
> > Additionally, even in longer sequences, the PGV frames exhibit a largely consistent pattern, with variations occurring mainly in areas of substantial motion like the hands. The uniformity in color distribution within each frame further underscores the spatial smoothness of the PGV representation.
> > This consistency underscores the effectiveness of our approach in maintaining spatial and temporal smoothness, even in dynamically moving sequences. Such spatio-temporal regularization facilitates efficient indexing for convolution operations in downstream tasks, such as sliding window techniques, and aids in feature learning.
> >
> > We sincerely hope that the above reply can fully address your raised concerns.

---

### Official Review · Reviewer_a2nk · 2023-10-30

**Soundness:** 2 fair
**Presentation:** 1 poor
**Contribution:** 2 fair
**Rating:** 5
**Confidence:** 2

**Summary:**

The paper introduces a novel approach to representing point cloud sequences, considering them as colored videos. This representation enables the application of 2D/3D convolution techniques, leading to improved performance in the task. Additionally, the author proposes specific micro-designs for temporal and spatial alignment of different frames within a point cloud sequence.

**Strengths:**

- Reformulate the task in aspect of input representation, which is remarkable and interesting.
- it is crucial to acknowledge that when applying convolutional kernels to point cloud data, achieving spatial consistency becomes a pivotal challenge

**Weaknesses:**

- The paper is a little hard to read, the method is not clarified clearly. How to optimize the network with EMD loss? How to obtain the point cloud from 'I'.
- How it will perform when deal with unseen data (o.o.d. data), since the author claimed they adopt a over-fitting manner to perform pc2pixel transforms.
- what's pre-defined 2d grids?
- what does "deforms a pre-defined 2D regular grid" mean.

**Questions:**

See above. I hope the author can provide more clarification in rebuttal, focusing on the detailed pipeline (like, how to represent the point cloud as image.)

---

> ### Author Response · Authors · 2023-11-18
> **Response to Reviewer a2nk**
>
> **Response to Reviewer a2nk**
>
> Thanks for your time and effort in reviewing our manuscript and  providing constructive comments. In the following, we address your concerns comprehensively.
>
> > ### Comment 1: How to optimize the network with EMD loss?
>
> **Response**: We refer the reviewer to our **overall response** for a comprehensive description of our PGV framework. Here, we provide a more detailed clarification of the content mentioned in the technical motivation in Section 3.
>
> Generally, we construct a network composed of 2D convolution (Conv2D) $f\_{{\psi}}(\cdot)$, with ${\psi}$ being the network parameters.
> The network $f\_{{\psi}}(\cdot)$ takes a pre-defined 2D regular grid $\mathbf{C} \in \mathbb{R}^{H \times W \times 3}$ as input.
> And we optimize $f\_{{\psi}}(\cdot)$ to over-fit each of $\\{\mathbf{P}_t\\}\_{t=0}^{T-1}$ independently as a 2D color image $\mathbf{I}_t = f\_{{\psi}_t}(\mathbf{C})$ by minimizing the EMD loss between $\mathbf{P}_t$ and the point cloud re-organized from $\mathbf{I}_t$, which means treating image pixels as point coordinates to form a point cloud $\mathbf{P}_t' \in \mathbb{R}^{(HW) \times 3}$, i.e., $\mathbf{P}_t'=\\{ \mathbf{I}_t[u,v,:] | 1 \leq u \leq H, 1 \leq v \leq W \\}=\\{f\_{{\psi}_t}(\mathbf{C})[u,v,:] | 1 \leq u \leq H, 1 \leq v \leq W \\}$. Here, $\mathbf{I}_t[u,v,:]$ represents the pixel values at position $(u, v)$ across all three color channels in $\mathbf{I}_t$.
> Then, the optimization using EMD loss can be formulated as
> $$
> \min\_{{\psi}_t} \sum\_{t=0}^{T-1} \mathcal{L}\_{\text{EMD}}\left(\mathbf{P}_t', \mathbf{P}_t \right)=\min\_{{\psi}_t} \sum\_{t=0}^{T-1} \mathcal{L}\_{\text{EMD}}\left( \\{ f\_{{\psi}_t}(\mathbf{C})[u,v,:]  | 1 \leq u \leq H, 1 \leq v \leq W \\}, \mathbf{P}_t \right)
> $$
>
> > ### Comment 2: How it will perform when deal with unseen data (o.o.d. data), since the author claimed they adopt a over-fitting manner to perform pc2pixel transforms.
>
> **Response**: We refer the reviewer to our **overall response** for a comprehensive description of our PGV framework. As addressed in our response to this reviewer's first question, during the representation stage (as seen in the middle of Fig. 2), the network is optimized with EMD loss.
>
> This optimization, which does not require any human-annotated labels, can operate on any 3D point cloud sequences with dimensions $T\times N\times 3$ and can be considered self-supervised. Therefore, the performance of the PGV representation stage remains unaffected, regardless of whether the data is unseen. We have applied our framework to various input sequences to generate their PGVs, as demonstrated in  Fig. 5(a), Fig.8 (b)-(c), Fig. A.7 and Fig. A.8.
>
> > ### Comment 3: what's pre-defined 2d grids? \& what does "deforms a pre-defined 2D regular grid" mean.
>
> **Response**: We have provided the details of the pre-defined 2D grid in the APPENDIX Section A.1.
> The pre-defined regular 2D grids are created by regularly sampling coordinates from the first slice of the 3D space $[0,~1]^3$, respectively. For a pre-defined 2D grid $\mathbf{C} \in \mathbb{R}^{H \times W \times 3}$, the values of each pixel coordinate $(u, v)$ are defined as $\mathbf{C}(u,v,1)=\frac{u-1}{H-1}$, $\mathbf{C}(u,v,2)=\frac{v-1}{W-1}$, and $\mathbf{C}(u,v,3) = 0$, where $1\leq u \leq H$ and $1\leq v \leq W$.
>
> Deforming a pre-defined 2D regular grid into a target 3D point cloud means utilizing the grid's three-channel values as 3D points coordinates, and minimizing the point set similarity between these coordinates and those of the target 3D point cloud. We implement this process as a reconstruction task, optimizing a network to input the grid's coordinates and supervising it by minimizing the Earth Mover's Distance (EMD) loss between the output coordinates and the target point cloud's coordinates.
>
> > ### Comment 4: provide more clarification in rebuttal, focusing on the detailed pipeline (like, how to represent the point cloud as image.)  \& How to obtain the point cloud from 'I'.
>
> **Response**: We refer the reviewer to our **overall response** for a comprehensive description of our PGV framework.
> To comprehend the transformation between the 3D point cloud and 2D image format, it is important to grasp the fundamental characteristics of these two formats.
> RGB images are essentially 5-dimensional signals, denoted as $(r, g, b, u, v)$, where the color information $(r, g, b)$ is indexed by regular grid coordinates $(u, v)$. In contrast, point cloud data is represented by spatial coordinates $(x, y, z)$, which provide both the geometric information and act as indirect cues for determining inter-point relationships, such as proximity and correspondence.
> For each frame of PGV, the transformation process between the point cloud and image format is executed by filling the spatial coordinates $(x,y,z)$ to the color channels $(r,g,b)$ at corresponding pixel locations $(u,v)$ on the 2D grid, and vice versa.

---

> ### Author Response · Authors · 2023-11-21
> **Looking Forward to Your Feedback. Thanks.**
>
> Dear **Reviewer a2nk**
>
> Thank you for dedicating your time and effort to reviewing our work. We have carefully considered and addressed all the concerns you raised in your review, as outlined in our response and reflected in the updated manuscript. As the Reviewer-Author discussion phase is nearing its conclusion, we eagerly await any further feedback from you. Should you have any additional questions, we would be delighted to provide detailed responses.

---

> > ### Author Response · Authors · 2023-11-23
> >
> > Dear Reviewer **a2nk**
> >
> > Thank you for dedicating your time and effort to reviewing our work and reading our responses. As the Reviewer-Author discussion phase is nearing its conclusion, we eagerly await any further feedback from you.
> >
> > Best regards,
> >
> > The authors

---

### Official Review · Reviewer_eyf9 · 2023-11-01

**Soundness:** 3 good
**Presentation:** 3 good
**Contribution:** 3 good
**Rating:** 3
**Confidence:** 4

**Summary:**

This paper proposes an approach to convert a 3D point cloud sequence into a 2D video and then utilize 2D convolution blocks to encode. They learn to obtain a grid by deforming a 2D grid to the 3D point cloud while minimizing the EMD loss between the original point cloud and the reconstructed version.

**Strengths:**

This paper is very novel as it tries something very difficult -- map a 3D point cloud into 2D. This is mathematically impossible for (even) many of the cases they presented in the paper, where the topological difference between the point cloud and the 2D domain precludes a diffeomorphism. The paper indeed made significant efforts to attempt to do this as well as possible.

**Weaknesses:**

Fundamentally I don't agree with this approach because mathematically it is impossible. Besides, the regular Conv3D is a poor way to handle temporal motion -- we don't use it even in regular 2D videos, e.g. for video object segmentation tasks, due to its incapability to handle motion. This paper has to make a lot of effort to get it to work only somewhat well, with a lot of efficiency limitations.

But putting personal opinions aside, the main problem is that this paper tested only on some fringe problems in 3D point clouds -- it would be more convincing if the evaluations were done on tasks such as semantic segmentation, object detection, instance segmentation, scene flow, point cloud generation etc. rather than the correspondence and upsampling tasks that were examined in this paper. Besides, they should compare with regular point cloud backbones such as PointTransformer, MinkowskiNet, PointConv etc. on both efficiency and speed. For the goal this paper is claiming -- superiority and generality, and "opens up new possibilities for point cloud sequences", I would think the current evaluation is far from enough.

**Questions:**

I believe this paper needs a major revision and redo their experiments. Hence I cannot support accepting it in its current form.

---

> ### Author Response · Authors · 2023-11-18
> **Response to Reviewer eyf9 (1/3)**
>
> **Response to Reviewer eyf9 (1/3)**
>
> Thanks for your time and effort in reviewing our manuscript and  providing constructive comments. In the following, we address your concerns comprehensively.
>
> > ### Comment 1: Fundamentally I don't agree with this approach because mathematically it is impossible.
>
> **Response**:
> First of all, we want to emphasize that our PGV representation has **fundamental differences** from traditional surface parameterization because we aim to achieve both spatial smoothness and temporal consistency in the resulting video. Traditional 3D surface parameterization focuses only on spatial smoothness.
> In the per-frame scenario, this problem is related to the 3D surface parameterization problem, as mentioned in both the technical motivation (Section 3) and the representation quality discussion (Section 6). We **fully acknowledge** the mathematical impossibility of achieving a perfect, distortion-free, and continuous bi-directional mapping of surface parameterization in certain scenarios, e.g., Figure A.9 discussed in Section 6. However, our deep learning approach is specifically designed to find an acceptable approximate solution in practical contexts. Leveraging the robust capabilities of deep learning, our method addresses the complex problem of 3D-to-2D mapping, despite theoretical limitations. **Our aim is to provide an approximation that is valuable for real-world applications rather than pursuing theoretical perfection.  More importantly, we experimentally demonstrated our PGV representation modality indeed brings significant advantages in various applications.**
>
> The approximation exploration process can also be depicted by our ablation study in Table 2. This study shows the transition from using FWSS only to address intra-frame smoothness, to using STJS only to address inter-frame consistency, and finally to using both FWSS and STJS to preserve both spatial smoothness and temporal consistency. This progression is evidenced by the evolution of the visualization (Figure 11) and the increasing correspondence ratio (Table 2). Besides, as discussed in Section 6 of our manuscript, we believe that the core of further optimizing this approximate solution lies in topological and geometric constraints, which may involve complex mathematical and algorithmic techniques such as adjusting mappings to preserve local features, using special mapping techniques (such as conformal mapping or isometric mapping).
>
> > ### Comment 2: Besides, the regular Conv3D is a poor way to handle temporal motion -- we don't use it even in regular 2D videos, e.g. for video object segmentation tasks, due to its incapability to handle motion.
>
> **Response**: We are concerned that the reviewer may **have misunderstood** our method. We would like to clarify that our method consists of two stages: (1) generating the PGV representation in a self-supervised manner, and (2) applying PGV to downstream applications, as demonstrated in Figure 2. The first stage utilizes modules that incorporate both 2D and 3D blocks.
>
> We use Conv2D/Conv3D blocks in the modules of **the representation stage** to ensure that the generated PGV preserves both spatial smoothness and temporal consistency, leveraging the inherent spectral bias property of neural networks [1]. The STJS module takes the results from the FWSS module with only coarse temporal consistency (Figure 11(a)), refining it into PGV with feasible temporal consistency (Figure 11(b)).
>
> [1] Rahaman et al. “On the spectral bias of neural networks,” in ICML 2019.
>
> Temporal motions (or trajectories) can be obtained by extracting values at the same pixel location along the temporal domain from PGV. The resulting PGV is a structured format compared to the original irregular 3D point cloud sequences. Additionally, we have the flexibility to employ either Conv2D or Conv3D to process the resulting PGV in downstream tasks.
>
> Regarding the applications stage, we have implemented various image-/video-based architectures, employing either 2D or 3D convolutions. For instance, in the task of future point cloud frame synthesis (referenced in Section 4), the regular structure of our PGV representation allows us to use two recent video processing models, namely FLAVR and SimVP, which utilize 3D and 2D convolutions, respectively. Detailed results of this application are presented in Section 5.3.

---

> ### Author Response · Authors · 2023-11-18
> **Response to Reviewer eyf9 (2/3)**
>
> **Response to Reviewer eyf9 (2/3)**
>
> > ### Comment 3: This paper has to make a lot of effort to get it to work only somewhat well, with a lot of efficiency limitations.
>
> **Response**: We have mentioned the issue of computational complexity in the discussion section of our paper (Section 6). Although our method is still limited by the efficiency of the EMD loss, we believe there is potential for improvement with the advancement of loss functions. Taking the popular NeRF method as an example, it initially faced high computational complexity upon its proposal, but subsequent NeRF-based works have improved its efficiency.
>
> From a different perspective, successfully generating PGV itself implies time savings, as the spatio-temporally regularized PGV representation described in our paper lacks a real-world ground truth and is even more challenging to achieve through human annotation. In other words, there are no readily available ground truth sequences that exhibit both spatial smoothness and temporal consistency. Even if it were possible to annotate them, creating such sequences through human annotation would require tedious efforts and consume a significant amount of time.
>  Additionally, the PGV representation can be used in an **offline** manner to **automatically** process irregular 3D point cloud sequences, and the resulting PGVs will be **permanently** stored, on which the conducted downstream tasks can greatly benefit in terms of efficiency, performance, and implementation.
>
> Last but not least, being an early attempt to apply spatio-temporal regularization to 3D point cloud sequences, we argue that solving the fundamental challenge of creating these regularized sequences from scratch is more crucial than addressing efficiency issues.
>
> > ### Comment 4: But putting personal opinions aside, the main problem is that this paper tested only on some fringe problems in 3D point clouds -- it would be more convincing if the evaluations were done on tasks such as semantic segmentation, object detection, instance segmentation, scene flow, point cloud generation etc. rather than the correspondence and upsampling tasks that were examined in this paper.
>
> **Response**: The reviewer has concerns about the reasonableness of our choices of specific downstream tasks for point cloud sequence processing. First of all, we need to clarify that there are no apparent theoretical/practical barriers to adapting our PGV representations to tasks like segmentation/detection/generation, because PGV essentially records the original point coordinates information, but with a regular structure. Intuitively, we just re-arrange the storage order of points to achieve spatial smoothness and temporal consistency. However, when determining the experimental setups of this paper, our major consideration factor lies in whether the performance of the chosen task can better reflect the representation quality of the generated PGVs. After all, the core of this work is to propose a new point cloud representation and demonstrate its potential, instead of pursuing state-of-the-art performances on popular benchmarks.
>
> Following such principle, we selected (1) sequence correspondence, (2) spatial up-sampling, and (3) frame forecasting, as our downstream tasks.
>
> - The first task can directly reflect the temporal consistency of PGVs.
>
> - The performance of the second task is influenced by the spatial smoothness of PGV frames.
>
> - The third task jointly reflects the PGV representation quality in both spatial and temporal domains.
>
> Therefore, it would be inappropriate to regard our chosen tasks as "some fringe problems". Instead, the three tasks are particularly selected to facilitate evaluating the PGV representation quality from different perspectives. Furthermore, it is worth arguing that correspondence has been a very critical problem and popular research topic over the years.
>
> In general, our chosen tasks actually pose more challenges than common high-level semantic understanding tasks, because their performances are more sensitive to the PGV representation quality. Besides, compared with label prediction tasks, traditional generative/reconstructive scenarios rely on distribution-based point cloud loss functions (CD, EMD) that are inefficient and hard to optimize, which can be naturally circumvented within PGV-based frameworks. Considering the capacity of a conference presentation, we believe that the superiority and potential of our PGV have already been effectively validated.

---

> ### Author Response · Authors · 2023-11-18
> **Response to Reviewer eyf9 (3/3)**
>
> **Response to Reviewer eyf9 (3/3)**
>
> > ### Comment 5: Besides, they should compare with regular point cloud backbones such as PointTransformer, MinkowskiNet, PointConv etc. on both efficiency and speed. For the goal this paper is claiming -- superiority and generality, and "opens up new possibilities for point cloud sequences", I would think the current evaluation is far from enough.
>
> **Response**: The selection of methods for comparison in our paper is driven by our pioneering approach towards generating a regular representation modality for 3D point cloud sequences, with a focus on geometrically meaningful properties: spatial smoothness and temporal consistency. Consequently, our comparisons concentrate on sequence-based and geometry-preserving learning methods, as opposed to general backbones applied to tasks with annotated labels.
>
> We have conducted comparisons with recent 3D point cloud sequence processing methods such as CorrNet3D (Fig. 5-6), Neuromorph (Fig. 5-6), and P4DTrans (Fig. 9), as they embody fundamental techniques in this field, including correspondence finding and spatial-temporal feature extraction. Furthermore, we evaluated prominent 3D point cloud processing methods for upsampling, such as MAFU and Neural Points (Fig. 7), because these approaches excel in preserving the geometric details of the resulting point clouds.
>
> Given the focus of our conference paper on 3D point cloud sequence processing, it is not feasible to examine all backbones for both single and sequential point clouds. Therefore, we have executed the most direct and relevant comparisons across a variety of tasks for both single and sequential point cloud processing methods.

---

> ### Author Response · Authors · 2023-11-21
> **Looking Forward to Your Feedback. Thanks.**
>
> Dear **Reviewer eyf9**,
>
> Thank you for dedicating your time and effort to reviewing our work and providing constructive comments. We have carefully considered and addressed all the concerns you raised in your review, as outlined in our response and reflected in the updated manuscript. As the Reviewer-Author discussion phase is nearing its conclusion, we eagerly await any further feedback from you. Should you have any additional questions, we would be delighted to provide detailed responses.

---

> > ### Author Response · Authors · 2023-11-23
> >
> > Dear  **Reviewer eyf9**,
> >
> > Thank you for dedicating your time and effort to reviewing our work and reading our responses. As the Reviewer-Author discussion phase is nearing its conclusion, we eagerly await any further feedback from you.
> >
> > Best regards,
> >
> > The authors

---

### Author Response · Authors · 2023-11-18
**Overall Response**

We are grateful for reviewers' recognition of the novelty and potential impact of our work. Before addressing the concern of each reviewer item by item, we re-emphasize that this paper introduces a new representation in the field of 3D point cloud sequence processing. This research area is still at its early development stage, and is essentially more challenging than static 3D point cloud processing due to the intricacy in both spatial and temporal domains.

As shown in Fig. 2 of the paper, our whole processing pipeline consists of **TWO** stages: (1) PGV representation and (2) downstream applications, where the essence lies in the first PGV representation stage. The input is an unstructured 3D point cloud sequence ($T\times N\times 3$) with $T$ frames, each of which contains $N$ 3D spatial points. The output is what we call Point Geometry Video (PGV), a well-structured video-format representation, with the dimension of $T\times H\times W\times 3$, where $N=H\times W$ and $H, W$ denote the height and width of the 2D grid plane. For each frame of PGV, the transformation between the point cloud and image format involves filling the coordinates channels $(x,y,z)$ into the color channels $(r,g,b)$ at the 2D grid's pixel location $(u,v)$, and vice versa.
In our representation phase, the primary focus is dedicated to the generation of PGV. This generation involves the employment of Conv2D/3D and the optimization of EMD loss.
In the downstream application phase, we apply the PGV to 3D point cloud sequence processing tasks. This application is significantly enhanced by the adoption of matured, video-based feature extraction techniques, coupled with the implementation of point-wise/pixel-wise Euclidean loss, thereby optimizing the performance across these diverse tasks.

To evaluate this, we applied PGV to point-based tasks rather than label prediction tasks. Typically, label prediction tasks are supervised with annotated labels, where point-wise categorical losses are already utilized, even for original irregular raw point cloud sequences. As a result, they do not adequately demonstrate the superiority of our structurization methods in preserving geometric details compared to point-based tasks such as correspondence, future frame synthesis, and sequence upsampling. These point-based tasks, which are self-supervised, previously depended on neighbor grouping and EMD/CD loss, struggling to preserve geometric details. This challenge is also one of the main gaps between 2D video and 3D point cloud sequence processing, which can be bridged by PGV.

**Last but not least**, the necessarily limited scope of this single manuscript does not fully reflect  the extensive efforts invested so far in our research agenda. To investigate this novel research paradigm, we have committed significant time and resources to methodological development, data preparation, and the crafting of various applications. The absence of ground truth further necessitates the creation of robust evaluation criteria and the development of suitable visualization techniques for both quantitative and qualitative analysis, representing a substantial workload. **Our presented method, while not yet perfect, has proven effective through extensive experiments, offering significant advantages and demonstrating considerable potential. We are confident that our research has introduced  fresh insights to the community and we sincerely hope that our contributions can be sufficiently recognized and properly valued.**

---

### Meta-Review · Area_Chair_NgTc · 2023-12-09

**Metareview:**

(a) This paper introduces an approach for transforming 3D point cloud sequences into 2D video representations, using a method termed PGV. It facilitates the application of 2D/3D convolution techniques to point cloud data, potentially enhancing performance in various tasks. The authors argue that their approach achieves both spatial smoothness and temporal consistency, which are critical in handling dynamic point cloud sequences. The paper presents experimental results on tasks like dense correspondence prediction, spatial up-sampling, and future point cloud frame synthesis.

(b) Strengths: The paper's primary strength lies in its innovative approach to representing 3D point cloud sequences. This approach is novel and insightful, particularly in its attempt to map complex 3D data into a 2D format, which has been a challenging area in the field. The methodology, as described, shows promise in facilitating the application of existing 2D and 3D convolution techniques to point cloud data. Additionally, the paper is well-written and presents its findings with clarity.

(c) Despite its strengths, the paper exhibits several weaknesses, as highlighted by the reviewers:

1. There are concerns regarding the limited scope of evaluation tasks. The paper focuses on specific tasks like correspondence and upsampling, which may not fully demonstrate the generality and superiority claimed by the authors. This problem is not well addressed in the rebuttal. (Reviewer eyf9)

2. Questions arise regarding the practical applicability of the method, especially in handling more complex/unseen data scenarios or diverse tasks. (Reviewer a2nk)

3. Concerns about the long-term performance and handling of incomplete or partial data are not fully addressed. (Reviewer BEST)

**Justification For Why Not Higher Score:**

1. There are concerns regarding the limited scope of evaluation tasks. The paper focuses on specific tasks like correspondence and upsampling, which may not fully demonstrate the generality and superiority claimed by the authors. This problem is not well addressed in the rebuttal. (Reviewer eyf9)

2. Questions arise regarding the practical applicability of the method, especially in handling more complex/unseen data scenarios or diverse tasks. (Reviewer a2nk)

3. Concerns about the long-term performance and handling of incomplete or partial data are not fully addressed. (Reviewer BEST)

**Justification For Why Not Lower Score:**

N/A

---

### Decision · Program_Chairs · 2024-01-16

Reject